# DIFFUSION POSTERIOR SAMPLING FOR GENERAL NOISY INVERSE PROBLEMS

**Hyungjin Chung**[*1,2]**, Jeongsol Kim**[*1]**, Michael T. Mccann**[2]**, Marc L. Klasky**[2] **& Jong Chul Ye**[1]
[1]KAIST,   [2] Los Alamos National Laboratory
{hj.chung, jeongsol, jong.ye}@kaist.ac.kr,   {mccann, mklasky}@lanl.gov

## ABSTRACT

Diffusion models have been recently studied as powerful generative inverse problem solvers, owing to their high quality reconstructions and the ease of combining existing iterative solvers. However, most works focus on solving simple linear inverse problems in noiseless settings, which significantly under-represents the complexity of real-world problems. In this work, we extend diffusion solvers to efficiently handle general noisy (non)linear inverse problems via approximation of the posterior sampling. Interestingly, the resulting posterior sampling scheme is a blended version of diffusion sampling with the manifold constrained gradient without a strict measurement consistency projection step, yielding a more desirable generative path in *noisy* settings compared to the previous studies. Our method demonstrates that diffusion models can incorporate various measurement noise statistics such as Gaussian and Poisson, and also efficiently handle noisy *nonlinear* inverse problems such as Fourier phase retrieval and non-uniform deblurring. Code is available at `https://github.com/DPS2022/diffusion-posterior-sampling`.

## 1 INTRODUCTION

Diffusion models learn the implicit prior of the underlying data distribution by matching the gradient of the log density (i.e. Stein score; $\nabla_{\boldsymbol{x}} \log p(\boldsymbol{x})$) (Song et al., 2021b). The prior can be leveraged when solving inverse problems, which aim to recover $\boldsymbol{x}$ from the measurement $\boldsymbol{y}$, related through the forward measurement operator $\mathcal{A}$ and the detector noise $\boldsymbol{n}$. When we know such forward models, one can incorporate the gradient of the log likelihood (i.e. $\nabla_{\boldsymbol{x}} \log p(\boldsymbol{y}|\boldsymbol{x})$) in order to sample from the posterior distribution $p(\boldsymbol{x}|\boldsymbol{y})$. While this looks straightforward, the likelihood term is in fact analytically intractable in terms of diffusion models, due to their dependence on time $t$. Due to its intractability, one often resorts to projections onto the measurement subspace (Song et al., 2021b; Chung et al., 2022b; Chung & Ye, 2022; Choi et al., 2021). However, the projection-type approach fails dramatically when 1) there is noise in the measurement, since the noise is typically amplified during the generative process due to the ill-posedness of the inverse problems; and 2) the measurement process is nonlinear.

One line of works that aim to solve noisy inverse problems run the diffusion in the spectral domain (Kawar et al., 2021; 2022) so that they can tie the noise in the measurement domain into the spectral domain via singular value decomposition (SVD). Nonetheless, the computation of SVD is costly and even prohibitive when the forward model gets more complex. For example, Kawar et al. (2022) only considered *seperable* Gaussian kernels for deblurring, since they were restricted to the family of inverse problems where they could effectively perform the SVD. Hence, the applicability of such methods is restricted, and it would be useful to devise a method to solve noisy inverse problems *without* the computation of SVD. Furthermore, while diffusion models were applied to various inverse problems including inpainting (Song et al., 2021b; Chung et al., 2022b; Kawar et al., 2022; Chung et al., 2022a), super-resolution (Choi et al., 2021; Chung et al., 2022b; Kawar et al., 2022), colorization (Song et al., 2021b; Kawar et al., 2022; Chung et al., 2022a), compressed-sensing MRI (CS-MRI) (Song et al., 2022; Chung & Ye, 2022; Chung et al., 2022b), computed tomography (CT) (Song et al., 2022; Chung et al., 2022a), etc., to our best knowledge, all works so far considered linear inverse problems only, and have not explored *nonlinear* inverse problems.

---

[*]Joint first authors

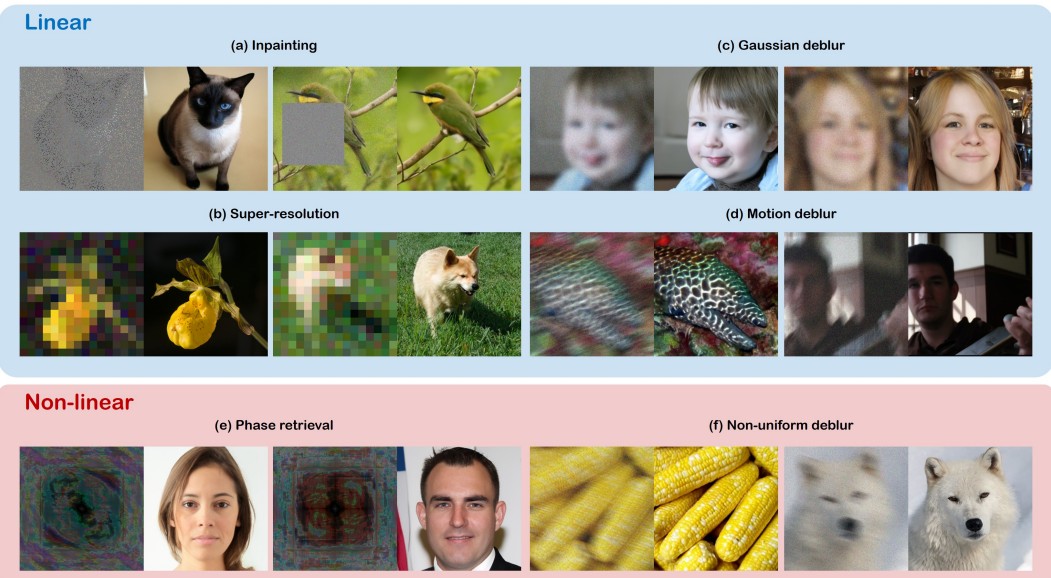

Figure 1: Solving noisy linear, and nonlinear inverse problems with diffusion models. Our reconstruction results (right) from the measurements (left) are shown.

In this work, we devise a method to circumvent the intractability of posterior sampling by diffusion models via a novel approximation, which can be generally applied to noisy inverse problems. Specifically, we show that our method can efficiently handle both the Gaussian and the Poisson measurement noise. Also, our framework easily extends to any nonlinear inverse problems, when the gradients can be obtained through automatic differentiation. We further reveal that a recently proposed method of manifold constrained gradients (MCG) (Chung et al., 2022a) is a special case of the proposed method when the measurement is *noiseless*. With a geometric interpretation, we further show that the proposed method is more likely to yield desirable sample paths in noisy setting than the previous approach (Chung et al., 2022a). In addition, the proposed method fully runs on the image domain rather than the spectral domain, thereby avoiding the computation of SVD for efficient implementation. With extensive experiments including various inverse problems—inpainting, super-resolution, (Gaussian/motion/non-uniform) deblurring, Fourier phase retrieval—we show that our method serves as a general framework for solving general noisy inverse problems with superior quality (Representative results shown in Fig. 1).

## 2 BACKGROUND

### 2.1 SCORE-BASED DIFFUSION MODELS

Diffusion models define the generative process as the *reverse* of the noising process. Specifically, Song et al. (2021b) defines the Itô stochastic differential equation (SDE) for the data noising process (i.e. forward SDE) $\boldsymbol{x}(t)$, $t \in [0, T]$, $\boldsymbol{x}(t) \in \mathbb{R}^d \, \forall t$ in the following form[1]

$$d\boldsymbol{x} = -\frac{\beta(t)}{2}\boldsymbol{x}dt + \sqrt{\beta(t)}d\boldsymbol{w}, \tag{1}$$

where $\beta(t) : \mathbb{R} \to \mathbb{R} > 0$ is the noise schedule of the process, typically taken to be monotonically increasing linear function of $t$ (Ho et al., 2020), and $\boldsymbol{w}$ is the standard $d-$dimensional Wiener process. The data distribution is defined when $t = 0$, i.e. $\boldsymbol{x}(0) \sim p_{\text{data}}$, and a simple, tractable distribution (e.g. isotropic Gaussian) is achieved when $t = T$, i.e. $\boldsymbol{x}(T) \sim \mathcal{N}(\boldsymbol{0}, \boldsymbol{I})$.

Our aim is to recover the data generating distribution starting from the tractable distribution, which can be achieved by writing down the corresponding reverse SDE of (1) (Anderson, 1982):

$$d\boldsymbol{x} = \left[-\frac{\beta(t)}{2}\boldsymbol{x} - \beta(t)\nabla_{\boldsymbol{x}_t} \log p_t(\boldsymbol{x}_t)\right]dt + \sqrt{\beta(t)}d\bar{\boldsymbol{w}}, \tag{2}$$

---

[1]In this work, we consider the variance preserving (VP) form of the SDE (Song et al., 2021b) which is equivalent to Denoising Diffusion Probabilistic Models (DDPM) (Ho et al., 2020).

where $dt$ corresponds to time running backward and $d\bar{w}$ to the standard Wiener process running backward. The drift function now depends on the time-dependent score function $\nabla_{\boldsymbol{x}_t} \log p_t(\boldsymbol{x}_t)$, which is approximated by a neural network $\boldsymbol{s}_\theta$ trained with denoising score matching (Vincent, 2011):

$$\theta^* = \arg\min_\theta \mathbb{E}_{t \sim U(\varepsilon,1), \boldsymbol{x}(t) \sim p(\boldsymbol{x}(t)|\boldsymbol{x}(0)), \boldsymbol{x}(0) \sim p_{\text{data}}} \left[ \|\boldsymbol{s}_\theta(\boldsymbol{x}(t), t) - \nabla_{\boldsymbol{x}_t} \log p(\boldsymbol{x}(t)|\boldsymbol{x}(0))\|_2^2 \right], \quad (3)$$

where $\varepsilon \simeq 0$ is a small positive constant. Once $\theta^*$ is acquired through (3), one can use the approximation $\nabla_{\boldsymbol{x}_t} \log p_t(\boldsymbol{x}_t) \simeq \boldsymbol{s}_{\theta^*}(\boldsymbol{x}_t, t)$ as a plug-in estimate[2] to replace the score function in (2). Discretization of (2) and solving using, e.g. Euler-Maruyama discretization, amounts to sampling from the data distribution $p(\boldsymbol{x})$, the goal of generative modeling.

Throughout the paper, we adopt the standard VP-SDE (i.e. ADM of Dhariwal & Nichol (2021) or Denoising Diffusion Probabilistic Models (DDPM) (Ho et al., 2020)), where the reverse diffusion variance which we denote by $\tilde{\sigma}(t)$ is learned as in Dhariwal & Nichol (2021). In discrete settings (e.g. in the algorithm) with $N$ bins, we define $\boldsymbol{x}_i \triangleq \boldsymbol{x}(tT/N)$, $\beta_i \triangleq \beta(tT/N)$, and subsequently $\alpha_i \triangleq 1 - \beta_i, \bar{\alpha}_i \triangleq \prod_{j=1}^i \alpha_i$ following Ho et al. (2020).

## 2.2 Inverse problem solving with diffusion models

For various scientific problems, we have a partial measurement $\boldsymbol{y}$ that is derived from $\boldsymbol{x}$. When the mapping $\boldsymbol{x} \mapsto \boldsymbol{y}$ is many-to-one, we arrive at an ill-posed inverse problem, where we cannot exactly retrieve $\boldsymbol{x}$. In the Bayesian framework, one utilizes $p(\boldsymbol{x})$ as the *prior*, and samples from the *posterior* $p(\boldsymbol{x}|\boldsymbol{y})$, where the relationship is formally established with the Bayes' rule: $p(\boldsymbol{x}|\boldsymbol{y}) = p(\boldsymbol{y}|\boldsymbol{x})p(\boldsymbol{x})/p(\boldsymbol{y})$. Leveraging the diffusion model as the prior, it is straightforward to modify (2) to arrive at the reverse diffusion sampler for sampling from the posterior distribution:

$$d\boldsymbol{x} = \left[ -\frac{\beta(t)}{2}\boldsymbol{x} - \beta(t)(\nabla_{\boldsymbol{x}_t} \log p_t(\boldsymbol{x}_t) + \nabla_{x_t} \log p_t(\boldsymbol{y}|\boldsymbol{x}_t)) \right] dt + \sqrt{\beta(t)}d\bar{\boldsymbol{w}}, \quad (4)$$

where we have used the fact that

$$\nabla_{\boldsymbol{x}_t} \log p_t(\boldsymbol{x}_t|\boldsymbol{y}) = \nabla_{\boldsymbol{x}_t} \log p_t(\boldsymbol{x}_t) + \nabla_{\boldsymbol{x}_t} \log p_t(\boldsymbol{y}|\boldsymbol{x}_t). \quad (5)$$

In (4), we have two terms that should be computed: the score function $\nabla_{\boldsymbol{x}_t} \log p_t(\boldsymbol{x}_t)$, and the likelihood $\nabla_{\boldsymbol{x}_t} \log p_t(\boldsymbol{y}|\boldsymbol{x}_t)$. To compute the former term involving $p_t(\boldsymbol{x})$, we can simply use the pre-trained score function $\boldsymbol{s}_{\theta^*}$. However, the latter term is hard to acquire in closed-form due to the dependence on the time $t$, as there only exists explicit dependence between $\boldsymbol{y}$ and $\boldsymbol{x}_0$.

Formally, the general form of the *forward* model[3] can be stated as

$$\boldsymbol{y} = \mathcal{A}(\boldsymbol{x}_0) + \boldsymbol{n}, \quad \boldsymbol{y}, \boldsymbol{n} \in \mathbb{R}^n, \boldsymbol{x} \in \mathbb{R}^d, \quad (6)$$

where $\mathcal{A}(\cdot): \mathbb{R}^d \mapsto \mathbb{R}^n$ is the forward measurement operator and $\boldsymbol{n}$ is the measurement noise. In the case of white Gaussian noise, $\boldsymbol{n} \sim \mathcal{N}(0, \sigma^2 \boldsymbol{I})$. In explicit form, $p(\boldsymbol{y}|\boldsymbol{x}_0) \sim \mathcal{N}(\boldsymbol{y}|\mathcal{A}(\boldsymbol{x}_0), \sigma^2 \boldsymbol{I})$. However, there does not exist explicit dependency between $\boldsymbol{y}$ and $\boldsymbol{x}_t$, as can be seen in the probabilistic graph from Fig. 2, where the blue dotted line remains unknown.

In order to circumvent using the likelihood term directly, alternating projections onto the measurement subspace is a widely used strategy (Song et al., 2021b; Chung & Ye, 2022; Chung et al., 2022b). Namely, one can disregard the likelihood term in (4), and first take an unconditional update with (2), and then take a projection step such that measurement consistency can be met, when assuming $\boldsymbol{n} \simeq 0$. Another line of work (Jalal et al., 2021) solves linear inverse problems where $\mathcal{A}(\boldsymbol{x}) \triangleq \boldsymbol{A}\boldsymbol{x}$ and utilizes an approximation $\nabla_{\boldsymbol{x}_t} \log p_t(\boldsymbol{y}|\boldsymbol{x}) \simeq \frac{\boldsymbol{A}^H(\boldsymbol{y}-\boldsymbol{A}\boldsymbol{x})}{\sigma^2}$, which is obtained when $\boldsymbol{n}$ is assumed to be Gaussian noise with variance $\sigma^2$. Nonetheless, the equation is only correct when $t = 0$, while being wrong at all other noise levels that are actually used in the generative process. The incorrectness is counteracted by a heuristic of assuming higher levels of noise as $t \to T$, such that $\nabla_{\boldsymbol{x}_t} \log p_t(\boldsymbol{y}|\boldsymbol{x}) \simeq \frac{\boldsymbol{A}^H(\boldsymbol{y}-\boldsymbol{A}\boldsymbol{x})}{\sigma^2 + \gamma_t^2}$, where $\{\gamma_t\}_{t=1}^T$ are hyperparameters. While both lines of works aim

---

[2]The approximation error comes from optimization/parameterization error of the neural network.

[3]To be precise, when we have signal-dependent noise model (e.g. Poisson), we cannot write the forward model with additive noise. We shall still write the forward model with additive noise for simplicity, and discuss which treatments are required when dealing with signal-dependent noise later in the paper.

to perform posterior sampling given the measurements and empirically work well on noiseless inverse problems, it should be noted that 1) they do not provide means to handle measurement noise, and 2) using such methods to solve nonlinear inverse problems either fails to work or is not straightforward to implement. The aim of this paper is to take a step toward a more general inverse problem solver, which can address noisy measurements and also scales effectively to nonlinear inverse problems.

## 3 DIFFUSION POSTERIOR SAMPLING (DPS)

### 3.1 APPROXIMATION OF THE LIKELIHOOD

Recall that no analytical formulation for $p(\boldsymbol{y}|\boldsymbol{x}_t)$ exists. In order to exploit the measurement model $p(\boldsymbol{y}|\boldsymbol{x}_0)$, we factorize $p(\boldsymbol{y}|\boldsymbol{x}_t)$ as follows:

$$p(\boldsymbol{y}|\boldsymbol{x}_t) = \int p(\boldsymbol{y}|\boldsymbol{x}_0, \boldsymbol{x}_t)p(\boldsymbol{x}_0|\boldsymbol{x}_t)d\boldsymbol{x}_0$$

$$= \int p(\boldsymbol{y}|\boldsymbol{x}_0)p(\boldsymbol{x}_0|\boldsymbol{x}_t)d\boldsymbol{x}_0, \qquad (7)$$

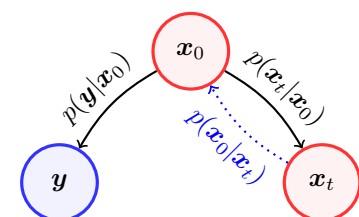

Figure 2: Probabilistic graph. Black solid line: tractable, blue dotted line: intractable in general.

where the second equality comes from that $\boldsymbol{y}$ and $\boldsymbol{x}_t$ are conditionally independent on $\boldsymbol{x}_0$, as shown in Fig. 2. Here, $p(\boldsymbol{x}_0|\boldsymbol{x}_t)$, as was shown with blue dotted lines in Fig. 2, is intractable in general. Note however, that for the case of diffusion models such as VP-SDE or DDPM, the forward diffusion can be simply represented by

$$\boldsymbol{x}_t = \sqrt{\bar{\alpha}(t)}\boldsymbol{x}_0 + \sqrt{1 - \bar{\alpha}(t)}\boldsymbol{z}, \qquad \boldsymbol{z} \sim \mathcal{N}(\boldsymbol{0}, \boldsymbol{I}), \qquad (8)$$

so that we can obtain the specialized representation of the posterior mean as shown in Proposition 1 through the Tweedie's approach (Efron, 2011; Kim & Ye, 2021). Detailed derivations can be found in Appendix A.

**Proposition 1.** *For the case of VP-SDE or DDPM sampling, $p(\boldsymbol{x}_0|\boldsymbol{x}_t)$ has the unique posterior mean at*

$$\hat{\boldsymbol{x}}_0 := \mathbb{E}[\boldsymbol{x}_0|\boldsymbol{x}_t] = \frac{1}{\sqrt{\bar{\alpha}(t)}}(\boldsymbol{x}_t + (1 - \bar{\alpha}(t))\nabla_{\boldsymbol{x}_t} \log p_t(\boldsymbol{x}_t)) \qquad (9)$$

**Remark 1.** *By replacing $\nabla_{\boldsymbol{x}_t} \log p(\boldsymbol{x}_t)$ in (9) with the score estimate $\boldsymbol{s}_{\theta^*}(\boldsymbol{x}_t)$, we can approximate the posterior mean from $p(\boldsymbol{x}_0|\boldsymbol{x}_t)$ as:*

$$\hat{\boldsymbol{x}}_0 \simeq \frac{1}{\sqrt{\bar{\alpha}(t)}}(\boldsymbol{x}_t + (1 - \bar{\alpha}(t))\boldsymbol{s}_{\theta^*}(\boldsymbol{x}_t, t)). \qquad (10)$$

*In fact, the result is closely related to the well established field of denoising. Concretely, consider the problem of retrieving the estimate of* clean $\boldsymbol{x}_0$ from the given Gaussian *noisy $\boldsymbol{x}_t$. A classic result of Tweedie's formula (Robbins, 1992; Stein, 1981; Efron, 2011; Kim & Ye, 2021) states that one can retrieve the empirical Bayes optimal posterior mean $\hat{\boldsymbol{x}}_0$ using the formula in (10).*

Given the posterior mean $\hat{\boldsymbol{x}}_0$ that can be efficiently computed at the intermediate steps, our proposal is to provide a tractable approximation for $p(\boldsymbol{y}|\boldsymbol{x}_t)$ such that one can use the surrogate function to maximize the likelihood—yielding approximate posterior sampling. Specifically, given the interpretation $p(\boldsymbol{y}|\boldsymbol{x}_t) = \mathbb{E}_{\boldsymbol{x}_0 \sim p(\boldsymbol{x}_0|\boldsymbol{x}_t)}[p(\boldsymbol{y}|\boldsymbol{x}_0)]$ from (7), we use the following approximation:

$$p(\boldsymbol{y}|\boldsymbol{x}_t) \simeq p(\boldsymbol{y}|\hat{\boldsymbol{x}}_0), \quad \text{where} \quad \hat{\boldsymbol{x}}_0 := \mathbb{E}[\boldsymbol{x}_0|\boldsymbol{x}_t] = \mathbb{E}_{\boldsymbol{x}_0 \sim p(\boldsymbol{x}_0|\boldsymbol{x}_t)}[\boldsymbol{x}_0] \qquad (11)$$

implying that the outer expectation of $p(\boldsymbol{y}|\boldsymbol{x}_0)$ over the posterior distribution is replaced with inner expectation of $\boldsymbol{x}_0$. In fact, this type of the approximation is closely related to the Jensen's inequality, so we need the following definition to quantify the approximation error:

**Definition 1** (Jensen gap (Gao et al., 2017; Simic, 2008)). *Let $\boldsymbol{x}$ be a random variable with distribution $p(\boldsymbol{x})$. For some function $f$ that may or may not be convex, the Jensen gap is defined as*

$$\mathcal{J}(f, \boldsymbol{x} \sim p(\boldsymbol{x})) = \mathbb{E}[f(\boldsymbol{x})] - f(\mathbb{E}[\boldsymbol{x}]), \qquad (12)$$

*where the expectation is taken over $p(\boldsymbol{x})$.*

| **Algorithm 1** DPS - Gaussian | **Algorithm 2** DPS - Poisson |
|---|---|
| **Require:** $N$, $\boldsymbol{y}$, $\{\zeta_i\}_{i=1}^N$, $\{\tilde{\sigma}_i\}_{i=1}^N$ | **Require:** $N$, $\boldsymbol{y}$, $\{\zeta_i\}_{i=1}^N$, $\{\tilde{\sigma}_i\}_{i=1}^N$ |
| 1: $\boldsymbol{x}_N \sim \mathcal{N}(\boldsymbol{0}, \boldsymbol{I})$ | 1: $\boldsymbol{x}_N \sim \mathcal{N}(\boldsymbol{0}, \boldsymbol{I})$ |
| 2: **for** $i = N - 1$ **to** 0 **do** | 2: **for** $i = N - 1$ **to** 0 **do** |
| 3: $\quad \hat{\boldsymbol{s}} \leftarrow \boldsymbol{s}_\theta(\boldsymbol{x}_i, i)$ | 3: $\quad \hat{\boldsymbol{s}} \leftarrow \boldsymbol{s}_\theta(\boldsymbol{x}_i, i)$ |
| 4: $\quad \hat{\boldsymbol{x}}_0 \leftarrow \frac{1}{\sqrt{\bar{\alpha}_i}}(\boldsymbol{x}_i + (1 - \bar{\alpha}_i)\hat{\boldsymbol{s}})$ | 4: $\quad \hat{\boldsymbol{x}}_0 \leftarrow \frac{1}{\sqrt{\bar{\alpha}_i}}(\boldsymbol{x}_i + (1 - \bar{\alpha}_i)\hat{\boldsymbol{s}})$ |
| 5: $\quad \boldsymbol{z} \sim \mathcal{N}(\boldsymbol{0}, \boldsymbol{I})$ | 5: $\quad \boldsymbol{z} \sim \mathcal{N}(\boldsymbol{0}, \boldsymbol{I})$ |
| 6: $\quad \boldsymbol{x}'_{i-1} \leftarrow \frac{\sqrt{\alpha_i}(1 - \bar{\alpha}_{i-1})}{1 - \bar{\alpha}_i}\boldsymbol{x}_i + \frac{\sqrt{\bar{\alpha}_{i-1}}\beta_i}{1 - \bar{\alpha}_i}\hat{\boldsymbol{x}}_0 + \tilde{\sigma}_i \boldsymbol{z}$ | 6: $\quad \boldsymbol{x}'_{i-1} \leftarrow \frac{\sqrt{\alpha_i}(1 - \bar{\alpha}_{i-1})}{1 - \bar{\alpha}_i}\boldsymbol{x}_i + \frac{\sqrt{\bar{\alpha}_{i-1}}\beta_i}{1 - \bar{\alpha}_i}\hat{\boldsymbol{x}}_0 + \tilde{\sigma}_i \boldsymbol{z}$ |
| 7: $\quad \textcolor{purple}{\boldsymbol{x}_{i-1} \leftarrow \boldsymbol{x}'_{i-1} - \zeta_i \nabla_{\boldsymbol{x}_i} \|\boldsymbol{y} - \mathcal{A}(\hat{\boldsymbol{x}}_0)\|_2^2}$ | 7: $\quad \textcolor{purple}{\boldsymbol{x}_{i-1} \leftarrow \boldsymbol{x}'_{i-1} - \zeta_i \nabla_{\boldsymbol{x}_i} \|\boldsymbol{y} - \mathcal{A}(\hat{\boldsymbol{x}}_0)\|_{\boldsymbol{\Lambda}}^2}$ |
| 8: **end for** | 8: **end for** |
| 9: **return** $\hat{\mathbf{x}}_0$ | 9: **return** $\hat{\mathbf{x}}_0$ |

The following theorem derives the closed-form upper bound of the Jensen gap for the inverse problem from (6) when $\boldsymbol{n} \sim \mathcal{N}(0, \sigma^2 \boldsymbol{I})$:

**Theorem 1.** *For the given measurement model (6) with $\boldsymbol{n} \sim \mathcal{N}(0, \sigma^2 \boldsymbol{I})$, we have*

$$p(\boldsymbol{y}|\boldsymbol{x}_t) \simeq p(\boldsymbol{y}|\hat{\boldsymbol{x}}_0), \tag{13}$$

*where the approximation error can be quantified with the Jensen gap, which is upper bounded by*

$$\mathcal{J} \leq \frac{d}{\sqrt{2\pi\sigma^2}} e^{-1/2\sigma^2} \|\nabla_{\boldsymbol{x}}\mathcal{A}(\boldsymbol{x})\| m_1, \tag{14}$$

*where $\|\nabla_{\boldsymbol{x}}\mathcal{A}(\boldsymbol{x})\| := \max_{\boldsymbol{x}} \|\nabla_{\boldsymbol{x}}\mathcal{A}(\boldsymbol{x})\|$ and $m_1 := \int \|\boldsymbol{x}_0 - \hat{\boldsymbol{x}}_0\| p(\boldsymbol{x}_0|\boldsymbol{x}_t) \, d\boldsymbol{x}_0$.*

**Remark 2.** *Note that $\|\nabla_{\boldsymbol{x}}\mathcal{A}(\boldsymbol{x})\|$ is finite in most of the inverse problems. This should not be confused with the ill-posedness of the inverse problems, which refers to the unboundedness of the inverse operator $\mathcal{A}^{-1}$. Accordingly, if $m_1$ is also finite (which is the case for most of the distribution in practice), the Jensen gap in Theorem 1 can approach to 0 as $\sigma \to \infty$, suggesting that the approximation error reduces with higher measurement noise. This may explain why our DPS works well for noisy inverse problems. In addition, although we have specified the measurement distribution to be Gaussian, we can also determine the Jensen gap for other measurement distributions (e.g. Poisson) in an analogous fashion.*

By leveraging the result of Theorem 1, we can use the approximate gradient of the log likelihood

$$\nabla_{\boldsymbol{x}_t} \log p(\boldsymbol{y}|\boldsymbol{x}_t) \simeq \nabla_{\boldsymbol{x}_t} \log p(\boldsymbol{y}|\hat{\boldsymbol{x}}_0), \tag{15}$$

where the latter is now analytically tractable, as the measurement distribution is given.

## 3.2 MODEL DEPENDENT LIKELIHOOD OF THE MEASUREMENT

Note that we may have different measurement models $p(\boldsymbol{y}|\boldsymbol{x}_0)$ for each application. Two of the most common cases in inverse problems are the Gaussian noise and the Poisson noise. Here, we explore how our diffusion posterior sampling described above can be adapted to each case.

**Gaussian noise.** The likelihood function takes the form

$$p(\boldsymbol{y}|\boldsymbol{x}_0) = \frac{1}{\sqrt{(2\pi)^n \sigma^{2n}}} \exp\left[-\frac{\|\boldsymbol{y} - \mathcal{A}(\boldsymbol{x}_0)\|_2^2}{2\sigma^2}\right],$$

where $n$ denotes the dimension of the measurement $\boldsymbol{y}$. By differentiating $p(\boldsymbol{y}|\boldsymbol{x}_t)$ with respect to $\boldsymbol{x}_t$, using Theorem 1 and (15), we get

$$\nabla_{\boldsymbol{x}_t} \log p(\boldsymbol{y}|\boldsymbol{x}_t) \simeq -\frac{1}{\sigma^2} \nabla_{\boldsymbol{x}_t} \|\boldsymbol{y} - \mathcal{A}(\hat{\boldsymbol{x}}_0(\boldsymbol{x}_t))\|_2^2$$

where we have explicitly denoted $\hat{\boldsymbol{x}}_0 := \hat{\boldsymbol{x}}_0(\boldsymbol{x}_t)$ to emphasize that $\hat{\boldsymbol{x}}_0$ is a function of $\boldsymbol{x}_t$. Consequently, taking the gradient $\nabla_{\boldsymbol{x}_t}$ amounts to taking the backpropagation through the network. Plugging in the result from Theorem 1 to (5) with the trained score function, we finally conclude that

$$\nabla_{\boldsymbol{x}_t} \log p_t(\boldsymbol{x}_t|\boldsymbol{y}) \simeq \boldsymbol{s}_{\theta^*}(\boldsymbol{x}_t, t) - \rho \nabla_{\boldsymbol{x}_t} \|\boldsymbol{y} - \mathcal{A}(\hat{\boldsymbol{x}}_0)\|_2^2, \tag{16}$$

where $\rho \triangleq 1/\sigma^2$ is set as the step size.

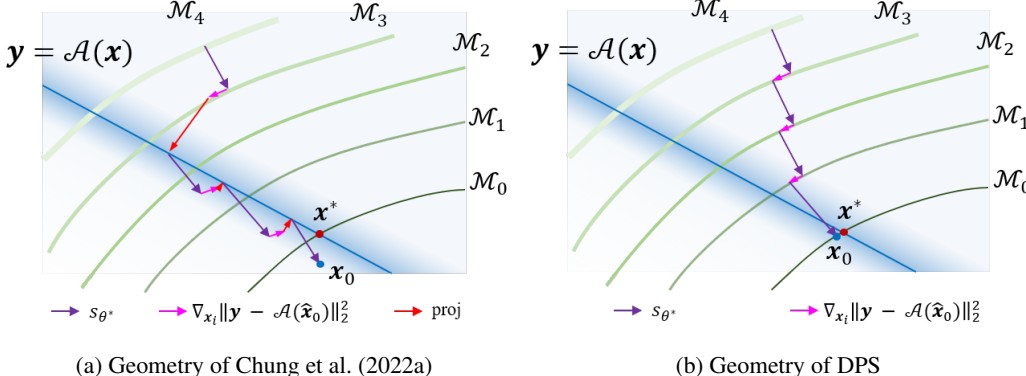

(a) Geometry of Chung et al. (2022a)  (b) Geometry of DPS

Figure 3: Conceptual illustration of the geometries of two different diffusion processes. Our method prevents the sample from falling off the generative manifolds when the measurements are noisy.

**Poisson noise.** The likelihood function for the Poisson measurements under the i.i.d. assumption is given as

$$p(\boldsymbol{y}|\boldsymbol{x}_0) = \prod_{j=1}^{n} \frac{[\mathcal{A}(\boldsymbol{x}_0)]_j^{\boldsymbol{y}_j} \exp\left[[-\mathcal{A}(\boldsymbol{x}_0)]_j\right]}{\boldsymbol{y}_j!}, \tag{17}$$

where $j$ indexes the measurement bin. In most cases where the measured values are not too small, the model can be approximated by a Gaussian distribution with very high accuracy[4]. Namely,

$$p(\boldsymbol{y}|\boldsymbol{x}_0) \to \prod_{j=1}^{n} \frac{1}{\sqrt{2\pi[\mathcal{A}(\boldsymbol{x}_0)]_j}} \exp\left(-\frac{(\boldsymbol{y}_j - [\mathcal{A}(\boldsymbol{x}_0)]_j)^2}{2[\mathcal{A}(\boldsymbol{x}_0)]_j}\right) \tag{18}$$

$$\simeq \prod_{j=1}^{n} \frac{1}{\sqrt{2\pi\boldsymbol{y}_j}} \exp\left(-\frac{(\boldsymbol{y}_j - [\mathcal{A}(\boldsymbol{x}_0)]_j)^2}{2\boldsymbol{y}_j}\right), \tag{19}$$

where we have used the standard approximation for the shot noise model $[\mathcal{A}(\boldsymbol{x}_0)]_j \simeq \boldsymbol{y}_j$ to arrive at the last equation (Kingston, 2013). Then, similar to the Gaussian case, by differentiation and the use of Theorem 1, we have that

$$\nabla_{\boldsymbol{x}_t} \log p(\boldsymbol{y}|\boldsymbol{x}_t) \simeq -\rho \nabla_{\boldsymbol{x}_t} \|\boldsymbol{y} - \mathcal{A}(\boldsymbol{x}_0)\|_{\boldsymbol{\Lambda}}^2, \quad [\boldsymbol{\Lambda}]_{ii} \triangleq 1/2\boldsymbol{y}_j, \tag{20}$$

where $\|\boldsymbol{a}\|_{\boldsymbol{\Lambda}}^2 \triangleq \boldsymbol{a}^T \boldsymbol{\Lambda} \boldsymbol{a}$, and we have included $\rho$ to define the step size as in the Gaussian case. We can summarize our strategy for each noise model as follows:

$$\nabla_{\boldsymbol{x}_t} \log p_t(\boldsymbol{x}_t|\boldsymbol{y}) \simeq s_{\theta^*}(\boldsymbol{x}_t, t) - \rho \nabla_{\boldsymbol{x}_t} \|\boldsymbol{y} - \mathcal{A}(\hat{\boldsymbol{x}}_0)\|_2^2 \qquad \text{(Gaussian)} \tag{21}$$

$$\nabla_{\boldsymbol{x}_t} \log p_t(\boldsymbol{x}_t|\boldsymbol{y}) \simeq s_{\theta^*}(\boldsymbol{x}_t, t) - \rho \nabla_{\boldsymbol{x}_t} \|\boldsymbol{y} - \mathcal{A}(\hat{\boldsymbol{x}}_0)\|_{\boldsymbol{\Lambda}}^2 \qquad \text{(Poisson)} \tag{22}$$

Incorporation of (16) or (21) into the usual ancestral sampling (Ho et al., 2020) steps leads to Algorithm 1,2[5]. Here, we name our algorithm **D**iffusion **P**osterior **S**ampling (DPS), as we construct our method in order to perform sampling from the posterior distribution. Notice that unlike prior methods that limit their applications to linear inverse problems $\mathcal{A}(\boldsymbol{x}) \triangleq \boldsymbol{A}\boldsymbol{x}$, our method is fully general in that we can also use nonlinear operators $\mathcal{A}(\cdot)$. To show that this is indeed the case, in experimental section we take the two notoriously hard nonlinear inverse problems: Fourier phase retrieval and non-uniform deblurring, and show that our method has very strong performance even in such challenging problem settings.

**Geometry of DPS and connection to manifold constrained gradient (MCG).** Interestingly, our method in the Gaussian measurement case corresponds to the manifold constrained gradient (MCG) step that was proposed in Chung et al. (2022a), when setting $\boldsymbol{W} = \boldsymbol{I}$ from Chung et al. (2022a).

---

[4]For $\boldsymbol{y}_j > 20$, the approximation holds within 1% of the error (Hubbard, 1970).

[5]In the discrete implementation, we instead use $\zeta_i$ to express the step size. From the experiments, we observe that taking $\zeta_i = \zeta'/\|\boldsymbol{y} - \mathcal{A}(\hat{\boldsymbol{x}}_0(\boldsymbol{x}_i))\|$, with $\zeta'$ set to constant, yields highly stable results. See Appendix D for details in the choice of step size.

| Method | SR ($\times 4$) | | Inpaint (box) | | Inpaint (random) | | Deblur (gauss) | | Deblur (motion) | |
|---|---|---|---|---|---|---|---|---|---|---|
| | FID ↓ | LPIPS ↓ | FID ↓ | LPIPS ↓ | FID ↓ | LPIPS ↓ | FID ↓ | LPIPS ↓ | FID ↓ | LPIPS ↓ |
| DPS (ours) | **39.35** | **0.214** | **33.12** | **0.168** | **21.19** | **0.212** | **44.05** | **0.257** | **39.92** | **0.242** |
| DDRM (Kawar et al., 2022) | 62.15 | 0.294 | 42.93 | 0.204 | 69.71 | 0.587 | 74.92 | 0.332 | - | - |
| MCG (Chung et al., 2022a) | 87.64 | 0.520 | 40.11 | 0.309 | 29.26 | 0.286 | 101.2 | 0.340 | 310.5 | 0.702 |
| PnP-ADMM (Chan et al., 2016) | 66.52 | 0.353 | 151.9 | 0.406 | 123.6 | 0.692 | 90.42 | 0.441 | 89.08 | 0.405 |
| Score-SDE (Song et al., 2021b) (ILVR (Choi et al., 2021)) | 96.72 | 0.563 | 60.06 | 0.331 | 76.54 | 0.612 | 109.0 | 0.403 | 292.2 | 0.657 |
| ADMM-TV | 110.6 | 0.428 | 68.94 | 0.322 | 181.5 | 0.463 | 186.7 | 0.507 | 152.3 | 0.508 |

Table 1: Quantitative evaluation (FID, LPIPS) of solving linear inverse problems on FFHQ 256×256-1k validation dataset. **Bold**: best, underline: second best.

However, Chung et al. (2022a) additionally performs projection onto the measurement subspace after the update step via (16), which can be thought of as corrections that are made for deviations from perfect data consistency. Borrowing the interpretation of diffusion models from Chung et al. (2022a), we compare the generative procedure geometrically. It was shown that in the context of diffusion models, a single denoising step via $s_{\theta*}$ corresponds to the orthogonal projection to the data manifold, and the gradient step $\nabla_{x_i} \|y - \mathcal{A}(\hat{x}_0)\|_2^2$ takes a step tangent to the current manifold. For *noisy* inverse problems, when taking projections on the measurement subspace after every gradient step as in Chung et al. (2022a), the sample may fall off the manifold, accumulate error, and arrive at the wrong solution, as can be seen in Fig. 3a, due to the overly imposing the data consistency that works only for *noiseless* measurement. On the other hand, our method without the projections on the measurement subspace is free from such drawbacks for noisy measurement (see Fig. 3b). Accordingly, while projections on the measurement subspace are useful for noiseless inverse problems that Chung et al. (2022a) tries to solve, they fail dramatically for noisy inverse problems that we try to solve. Finally, when used together with the projection steps on the measurement subspace, it was shown that choosing different $W$ for different applications was necessary for MCG, whereas our method is free from such heuristics.

## 4 EXPERIMENTS

**Experimental setup.** We test our experiment on two datasets that have diverging characteristic - FFHQ 256×256 (Karras et al., 2019), and Imagenet 256×256 (Deng et al., 2009), on 1k validation images each. The pre-trained diffusion model for ImageNet was taken from Dhariwal & Nichol (2021) and was used directly without finetuning for specific tasks. The diffusion model for FFHQ was trained from scratch using 49k training data (to exclude 1k validation set) for 1M steps. All images are normalized to the range $[0, 1]$. Forward measurement operators are specified as follows: (i) For box-type inpainting, we mask out 128×128 box region following Chung et al. (2022a), and for random-type we mask out 92% of the total pixels (all RGB channels). (ii) For super-resolution, bicubic downsampling is performed. (iii) Gaussian blur kernel has size 61×61 with standard deviation of 3.0, and motion blur is randomly generated with the code[6], with size $61 \times 61$ and intensity value 0.5. The kernels are convolved with the ground truth image to produce the measurement. (iv) For phase retrieval, Fourier transform is performed to the image, and only the Fourier magnitude is taken as the measurement. (v) For nonlinear deblurring, we leverage the neural network approximated forward model as in Tran et al. (2021). All Gaussian noise is added to the measurement domain with $\sigma = 0.05$. Poisson noise level is set to $\lambda = 1.0$. More details including the hyper-parameters can be found in Appendix B,D.

We perform comparison with the following methods: Denoising diffusion restoration models (DDRM) (Kawar et al., 2022), manifold constrained gradients (MCG) (Chung et al., 2022a), Plug-and-play alternating direction method of multipliers (PnP-ADMM) (Chan et al., 2016) using DnCNN Zhang et al. (2017) in place of proximal mappings, total-variation (TV) sparsity regularized optimization method (ADMM-TV), and Score-SDE (Song et al., 2021b). Note that Song et al. (2021b) only proposes a method for inpainting, and not for gen-

| Method | FID ↓ | LPIPS ↓ |
|---|---|---|
| DPS(ours) | **55.61** | **0.399** |
| OSS | 137.7 | 0.635 |
| HIO | 96.40 | 0.542 |
| ER | 214.1 | 0.738 |

Table 3: Quantitative evaluation of the Phase Retrieval task (FFHQ).

[6]https://github.com/LeviBorodenko/motionblur

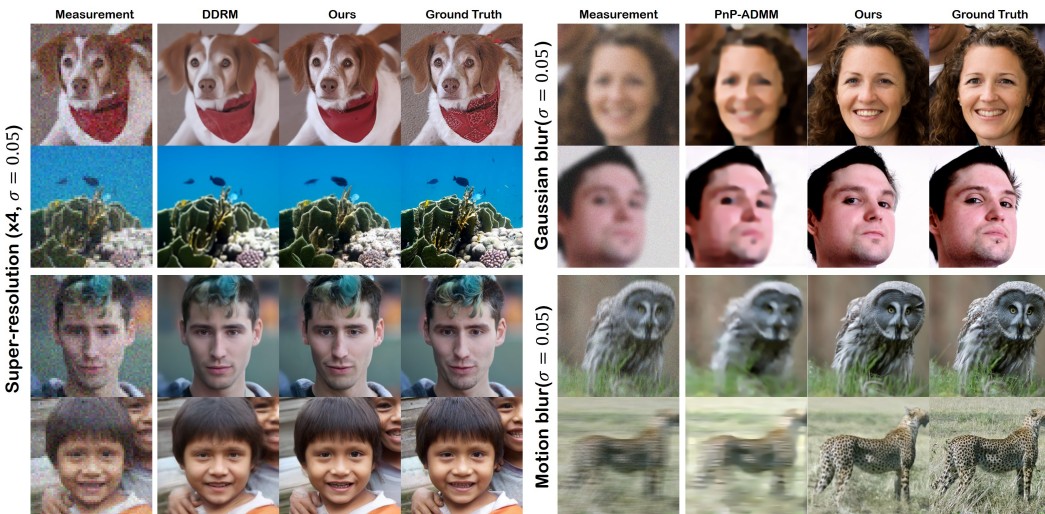

Figure 4: Results on solving linear inverse problems with Gaussian noise ($\sigma = 0.05$).

| Method | SR ($\times 4$) | | Inpaint (box) | | Inpaint (random) | | Deblur (gauss) | | Deblur (motion) | |
|---|---|---|---|---|---|---|---|---|---|---|
| | FID↓ | LPIPS↓ | FID↓ | LPIPS↓ | FID↓ | LPIPS↓ | FID↓ | LPIPS↓ | FID↓ | LPIPS↓ |
| DPS (ours) | **50.66** | **0.337** | **38.82** | 0.262 | **35.87** | **0.303** | **62.72** | 0.444 | **56.08** | **0.389** |
| DDRM (Kawar et al., 2022) | 59.57 | 0.339 | 45.95 | **0.245** | 114.9 | 0.665 | 63.02 | **0.427** | - | - |
| MCG (Chung et al., 2022a) | 144.5 | 0.637 | 39.74 | 0.330 | 39.19 | 0.414 | 95.04 | 0.550 | 186.9 | 0.758 |
| PnP-ADMM (Chan et al., 2016) | 97.27 | 0.433 | 78.24 | 0.367 | 114.7 | 0.677 | 100.6 | 0.519 | 89.76 | 0.483 |
| Score-SDE (Song et al., 2021b) (ILVR (Choi et al., 2021)) | 170.7 | 0.701 | 54.07 | 0.354 | 127.1 | 0.659 | 120.3 | 0.667 | 98.25 | 0.591 |
| ADMM-TV | 130.9 | 0.523 | 87.69 | 0.319 | 189.3 | 0.510 | 155.7 | 0.588 | 138.8 | 0.525 |

Table 2: Quantitative evaluation (FID, LPIPS) of solving linear inverse problems on ImageNet 256×256-1k validation dataset. **Bold**: best, underline: second best.

eral inverse problems. However, the methodology of iteratively applying projections onto convex sets (POCS) was applied in the same way for super-resolution in iterative latent variable refinement (ILVR) (Choi et al., 2021), and more generally to linear inverse problems in Chung et al. (2022b); thus we simply refer to these methods as score-SDE henceforth.For a fair comparison, we used the same score function for all the different methods that are based on diffusion (i.e. DPS, DDRM, MCG, score-SDE).

For phase retrieval, we compare with three strong baselines that are considered standards: oversampling smoothness (OSS) (Rodriguez et al., 2013), Hybrid input-output (HIO) (Fienup & Dainty, 1987), and error reduction (ER) algorithm (Fienup, 1982). For nonlinear deblurring, we compare against the prior arts: blur kernel space (BKS) - styleGAN2 (Tran et al., 2021), based on GAN priors, blur kernel space (BKS) - generic (Tran et al., 2021), based on Hyper-Laplacian priors, and MCG.

Further experimental details are provided in Appendix D. For quantitative comparison, we focus on the following two widely-used perceptual metrics - Fréchet Inception Distance (FID), and Learned Perceptual Image Patch Similarity (LPIPS) distance, with further evaluation with standard metrics: peak signal-to-noise-ratio (PSNR), and structural similarity index (SSIM) provided in Appendix E.

**Noisy linear inverse problems.** We first test our method on diverse linear inverse problems with Gaussian measurement noises. The quantitative results shown in Tables 1,2 illustrate that the proposed method outperforms all the other comparison methods by large margins. Particularly, MCG and Score-SDE (or ILVR) are methods that rely on projections on the measurement subspace, where the generative process is controlled such that the measurement consistency is *perfectly* met. While this

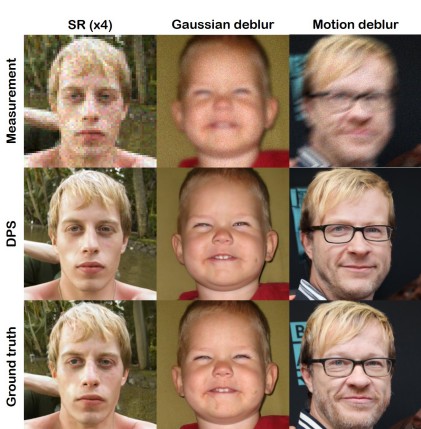

Figure 5: Results on solving linear inverse problems with Poisson noise ($\lambda = 1.0$).

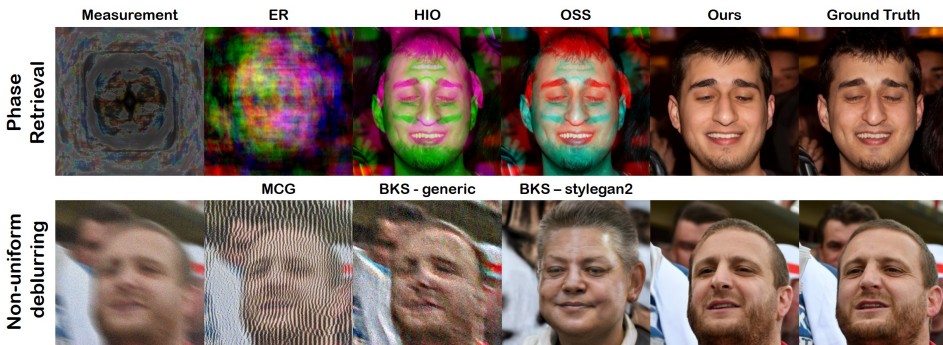

Figure 6: Results on solving nonlinear inverse problems with Gaussian noise ($\sigma = 0.05$).

is useful for noiseless (or negligible noise) problems, in the case where we cannot ignore noise, the solutions overfit to the corrupted measurement (for further discussion, see Appendix C.1). In Fig. 4, we specifically compare our methods with DDRM and PnP-ADMM, which are two methods that are known to be robust to measurement noise. Our method is able to provide high-quality reconstructions that are crisp and realistic on all tasks. On the other hand, we see that DDRM performs poorly on image inpainting tasks where the dimensionality of the measurements are very low, and tend to produce blurrier results on both SR, and deblurring tasks. We further note that DDRM relies on SVD, and hence is only able to solve problems where the forward measurement matrix can be efficiently implemented (e.g. separable kernel in the case of deblurring). Hence, while one can solve Gaussian deblurring, one cannot solve problems such as motion deblur, where the point spread function (PSF) is much more complex. Contrarily, our method is not restricted by such conditions, and can be always used regardless of the complexity. The results of the Poisson noisy linear inverse problems are presented in Fig. 5. Consistent with the Gaussian case, DPS is capable of producing high quality reconstructions that closely mimic the ground truth. From the experiments, we further observe that the weighted least squares method adopted in Algorithm 2 works best compared to other choices that can be made for Poisson inverse problems (for further analysis, see Appendix C.4).

**Nonlinear inverse problems.** We show the quantitative results of phase retrieval in Table 3, and the results of nonlinear deblurring in Table 4. Representative results are illustrated in Fig. 6.

We first observe that the proposed method is capable of highly accurate reconstruction for the given phase retrieval problem, capturing most of the high frequency details. However, we also observe that we do not *always* get high quality reconstructions. In fact, due to the non-uniqueness of the phase-retrieval under some conditions, widely used methods such as HIO are also dependent on the initializations (Fienup, 1978), and hence it is considered standard practice to first generate multiple reconstructions, and take the best sample. Following this, when reporting our quantitative metrics, we generate 4 different samples for all the methods, and report the metric based on the best samples. We see that DPS

| Method | FID ↓ | LPIPS ↓ |
|---|---|---|
| DPS(ours) | **41.86** | 0.278 |
| BKS-styleGAN2 | 63.18 | 0.407 |
| BKS-generic | 141.0 | 0.640 |
| MCG | 180.1 | 0.695 |

Table 4: Quantitative evaluation of the non-uniform deblurring task (FFHQ).

outperforms other methods by a large margin. For the case of nonlinear deblurring, we again see that our method performs the best, producing highly realistic samples. BKS-styleGAN2 (Tran et al., 2021) leverages GAN prior and hence generates feasible human faces, but heavily distorts the identity. BKS-generic utilizes the Hyper-Laplacian prior (Krishnan & Fergus, 2009), but is unable to remove artifacts and noise properly. MCG amplifies noise in a similar way that was discussed in Fig. 7.

## 5 CONCLUSION

In this paper, we proposed diffusion posterior sampling (DPS) strategy for solving general noisy (both signal dependent/independent) inverse problems in imaging. Our method is versatile in that it can also be used for highly noisy and nonlinear inverse problems. Extensive experiments show that the proposed method outperforms existing state-of-the-art by large margins, and also covers the widest range of problems.

ACKNOWLEDGMENTS

This work was supported by the National Research Foundation of Korea under Grant NRF-2020R1A2B5B03001980, by the Korea Medical Device Development Fund grant funded by the Korea government (the Ministry of Science and ICT, the Ministry of Trade, Industry and Energy, the Ministry of Health & Welfare, the Ministry of Food and Drug Safety) (Project Number: 1711137899, KMDF_PR_20200901_0015), and by the KAIST Key Research Institute (Interdisciplinary Research Group) Project.

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

## A    PROOFS

**Lemma 1** (Tweedie's formula). *Let $p(\boldsymbol{y}|\boldsymbol{\eta})$ belong to the exponential family distribution*

$$p(\boldsymbol{y}|\boldsymbol{\eta}) = p_0(\boldsymbol{y}) \exp(\boldsymbol{\eta}^\top T(\boldsymbol{y}) - \varphi(\boldsymbol{\eta})), \tag{23}$$

*where $\boldsymbol{\eta}$ is the canonical vector of the family, $T(\boldsymbol{y})$ is some function of $\boldsymbol{y}$, and $\varphi(\boldsymbol{\eta})$ is the cumulant generation function which normalizes the density, and $p_0(\boldsymbol{y})$ is the density up to the scale factor when $\boldsymbol{\eta} = \boldsymbol{0}$. Then, the posterior mean $\hat{\boldsymbol{\eta}} := \mathbb{E}[\boldsymbol{\eta}|\boldsymbol{y}]$ should satisfy*

$$(\nabla_{\boldsymbol{y}} T(\boldsymbol{y}))^\top \hat{\boldsymbol{\eta}} = \nabla_{\boldsymbol{y}} \log p(\boldsymbol{y}) - \nabla_{\boldsymbol{y}} \log p_0(\boldsymbol{y}) \tag{24}$$

*Proof.* Marginal distribution $p(\boldsymbol{y})$ could be expressed as

$$p(\boldsymbol{y}) = \int p(\boldsymbol{y}|\boldsymbol{\eta})p(\boldsymbol{\eta})d\boldsymbol{\eta} \tag{25}$$

$$= \int p_0(\boldsymbol{y}) \exp\left(\boldsymbol{\eta}^\top T(\boldsymbol{y}) - \varphi(\boldsymbol{\eta})\right)p(\boldsymbol{\eta})d\boldsymbol{\eta}. \tag{26}$$

Then, the derivative of the marginal distribution $p(\boldsymbol{y})$ with respect to $\boldsymbol{y}$ becomes

$$\nabla_{\boldsymbol{y}} p(\boldsymbol{y}) = \nabla_y p_0(\boldsymbol{y}) \int \exp\left(\boldsymbol{\eta}^\top T(\boldsymbol{y}) - \varphi(\boldsymbol{\eta})\right)p(\boldsymbol{\eta})d\boldsymbol{\eta} + \int (\nabla_{\boldsymbol{y}} T(\boldsymbol{y}))^\top \boldsymbol{\eta} p_0(\boldsymbol{y}) \exp\left(\boldsymbol{\eta}^\top T(\boldsymbol{y}) - \varphi(\boldsymbol{\eta})\right)p(\boldsymbol{\eta})d\boldsymbol{\eta}$$

$$= \frac{\nabla_y p_0(\boldsymbol{y})}{p_0(\boldsymbol{y})} \int p(\boldsymbol{y}|\boldsymbol{\eta})p(\boldsymbol{\eta})d\boldsymbol{\eta} + (\nabla_{\boldsymbol{y}} T(\boldsymbol{y}))^\top \int \boldsymbol{\eta} p(\boldsymbol{y}|\boldsymbol{\eta})p(\boldsymbol{\eta})d\boldsymbol{\eta}$$

$$= \frac{\nabla_y p_0(\boldsymbol{y})}{p_0(\boldsymbol{y})} p(\boldsymbol{y}) + (\nabla_{\boldsymbol{y}} T(\boldsymbol{y}))^\top \int \boldsymbol{\eta} p(\boldsymbol{y}, \boldsymbol{\eta})d\boldsymbol{\eta}$$

Therefore,

$$\frac{\nabla_y p(\boldsymbol{y})}{p(\boldsymbol{y})} = \frac{\nabla_y p_0(\boldsymbol{y})}{p_0(\boldsymbol{y})} + (\nabla_{\boldsymbol{y}} T(\boldsymbol{y}))^\top \int \boldsymbol{\eta} p(\boldsymbol{\eta}|\boldsymbol{y})d\boldsymbol{\eta} \tag{27}$$

which is equivalent to

$$(\nabla_{\boldsymbol{y}} T(\boldsymbol{y}))^\top \mathbb{E}[\boldsymbol{\eta}|\boldsymbol{y}] = \nabla_{\boldsymbol{y}} \log p(\boldsymbol{y}) - \nabla_{\boldsymbol{y}} \log p_0(\boldsymbol{y}) \tag{28}$$

This concludes the proof. □

**Proposition 1.** *For the case of VP-SDE or DDPM sampling, $p(\boldsymbol{x}_0|\boldsymbol{x}_t)$ has the unique posterior mean at*

$$\hat{\boldsymbol{x}}_0 := \mathbb{E}[\boldsymbol{x}_0|\boldsymbol{x}_t] = \frac{1}{\sqrt{\bar{\alpha}(t)}}(\boldsymbol{x}_t + (1 - \bar{\alpha}(t))\nabla_{\boldsymbol{x}_t} \log p_t(\boldsymbol{x}_t)) \tag{9}$$

*Proof.* For the case of VP-SDE and DDPM forward sampling in (8), we have

$$p(\boldsymbol{x}_t|\boldsymbol{x}_0) = \frac{1}{(2\pi(1 - \bar{\alpha}(t)))^{d/2}} \exp\left(-\frac{\|\boldsymbol{x}_t - \sqrt{\bar{\alpha}(t)}\boldsymbol{x}_0\|^2}{2(1 - \bar{\alpha}(t))}\right), \tag{29}$$

which is a Gaussian distribution. The corresponding canonical decomposition is then given by

$$p(\boldsymbol{x}_t|\boldsymbol{x}_0) = p_0(\boldsymbol{x}_t) \exp\left(\boldsymbol{x}_0^\top T(\boldsymbol{x}_t) - \varphi(\boldsymbol{x}_0)\right), \tag{30}$$

where

$$p_0(\boldsymbol{x}_t) := \frac{1}{(2\pi(1 - \bar{\alpha}(t)))^{d/2}} \exp\left(-\frac{\|\boldsymbol{x}_t\|^2}{2(1 - \bar{\alpha}(t))}\right)$$

$$T(\boldsymbol{x}_t) := \frac{\sqrt{\bar{\alpha}(t)}}{1 - \bar{\alpha}(t)}\boldsymbol{x}_t$$

$$\varphi(\boldsymbol{x}_0) := \frac{\bar{\alpha}(t)\|\boldsymbol{x}_0\|^2}{2(1 - \bar{\alpha}(t))}$$

Therefore, using (24), we have

$$\frac{\sqrt{\bar{\alpha}(t)}}{1 - \bar{\alpha}(t)} \hat{\boldsymbol{x}}_0 = \nabla_{\boldsymbol{x}_t} \log p_t(\boldsymbol{x}_t) + \frac{1}{1 - \bar{\alpha}(t)} \boldsymbol{x}_t$$

which leads to

$$\hat{\boldsymbol{x}}_0 = \frac{1}{\sqrt{\bar{\alpha}(t)}} \left( \boldsymbol{x}_t + (1 - \bar{\alpha}(t)) \nabla_{\boldsymbol{x}_t} \log p_t(\boldsymbol{x}_t) \right) \tag{31}$$

This concludes the proof. $\qquad\square$

**Proposition 2** (Jensen gap upper bound (Gao et al., 2017))**.** *Define the absolute cenetered moment as* $m_p := \sqrt[p]{\mathbb{E}[|X - \mu|^p]}$*, and the mean as* $\mu = \mathbb{E}[X]$*. Assume that for* $\alpha > 0$*, there exists a positive number* $K$ *such that for any* $x \in \mathbb{R}, |f(x) - f(\mu)| \leq K|x - \mu|^\alpha$*. Then,*

$$|\mathbb{E}[f(X) - f(\mathbb{E}[X])]| \leq \int |f(X) - f(\mu)| dp(X) \tag{32}$$

$$\leq K \int |x - \mu|^\alpha dp(X) \leq M m_\alpha^\alpha. \tag{33}$$

**Lemma 2.** *Let* $\phi(\cdot)$ *be a univariate Gaussian density function with mean* $\mu$ *and variance* $\sigma^2$*. There exists a constant* $L$ *such that* $\forall x, y \in \mathbb{R}$*,*

$$|\phi(x) - \phi(y)| \leq L|x - y|, \tag{34}$$

*where* $L = \frac{1}{\sqrt{2\pi\sigma^2}} \exp\left(-\frac{1}{2\sigma^2}\right)$*.*

*Proof.* As $\phi'$ is continuous and bounded, we use the mean value theorem to get

$$\forall (x, y) \in \mathbb{R}^2, |\phi(x) - \phi(y)| \leq \|\phi'\|_\infty |x - y|. \tag{35}$$

Since $L$ is the minimal value for (34), we have that $L \leq \|\phi'\|_\infty$. Taking the limit $y \to x$ gives $|\phi'(x)| \leq L$, and thus $\|\phi'\|_\infty \leq L$. Hence

$$L = \|\phi'\|_\infty = \| - \frac{x - \mu}{\sigma^2} \phi(x) \|_\infty. \tag{36}$$

Since the derivative of $\phi'$ is given as

$$\phi''(x) = \sigma^{-2}(1 - \sigma^{-2}(x - \mu)^2)\phi(x), \tag{37}$$

and the maximum is attained when $x = 1 \pm \sigma^2 \mu$, we have

$$L = \|\phi'\|_\infty = \frac{e^{-1/2\sigma^2}}{\sqrt{2\pi\sigma^2}} \tag{38}$$

$\qquad\square$

**Lemma 3.** *Let* $\phi(\cdot)$ *be an isotropic multivariate Gaussian density function with mean* $\boldsymbol{\mu}$ *and variance* $\sigma^2 \boldsymbol{I}$*. There exists a constant* $L$ *such that* $\forall \boldsymbol{x}, \boldsymbol{y} \in \mathbb{R}^d$*,*

$$\|\phi(\boldsymbol{x}) - \phi(\boldsymbol{y})\| \leq L\|\boldsymbol{x} - \boldsymbol{y}\|, \tag{39}$$

*where* $L = \frac{d}{\sqrt{2\pi\sigma^2}} e^{-1/2\sigma^2}$*.*

*Proof.*

$$\|\phi(\boldsymbol{x}) - \phi(\boldsymbol{y})\| \leq \max_{\boldsymbol{z}} \|\nabla_{\boldsymbol{z}} \phi(\boldsymbol{z})\| \cdot \|\boldsymbol{x} - \boldsymbol{y}\| \tag{40}$$

$$= \underbrace{\frac{d}{\sqrt{2\pi\sigma^2}} \exp\left(-\frac{1}{2\sigma^2}\right)}_{L} \cdot \|\boldsymbol{x} - \boldsymbol{y}\| \tag{41}$$

where the second inequality comes from that each element of $\nabla_{\boldsymbol{z}} \phi(\boldsymbol{z})$ is bounded by $\frac{1}{\sqrt{2\pi\sigma^2}} \exp\left(-\frac{1}{2\sigma^2}\right)$. $\qquad\square$

**Theorem 1.** *For the given measurement model (6) with $\boldsymbol{n} \sim \mathcal{N}(0, \sigma^2 \boldsymbol{I})$, we have*

$$p(\boldsymbol{y}|\boldsymbol{x}_t) \simeq p(\boldsymbol{y}|\hat{\boldsymbol{x}}_0), \tag{13}$$

*where the approximation error can be quantified with the Jensen gap, which is upper bounded by*

$$\mathcal{J} \leq \frac{d}{\sqrt{2\pi\sigma^2}} e^{-1/2\sigma^2} \|\nabla_{\boldsymbol{x}} \mathcal{A}(\boldsymbol{x})\| m_1, \tag{14}$$

*where $\|\nabla_{\boldsymbol{x}} \mathcal{A}(\boldsymbol{x})\| := \max_{\boldsymbol{x}} \|\nabla_{\boldsymbol{x}} \mathcal{A}(\boldsymbol{x})\|$ and $m_1 := \int \|\boldsymbol{x}_0 - \hat{\boldsymbol{x}}_0\| p(\boldsymbol{x}_0|\boldsymbol{x}_t) \, d\boldsymbol{x}_0$.*

*Proof.*

$$p(\boldsymbol{y}|\boldsymbol{x}_t) = \int p(\boldsymbol{y}|\boldsymbol{x}_0) p(\boldsymbol{x}_0|\boldsymbol{x}_t) d\boldsymbol{x}_0 \tag{42}$$

$$= \mathbb{E}_{\boldsymbol{x}_0 \sim p(\boldsymbol{x}_0|\boldsymbol{x}_t)}[f(\boldsymbol{x}_0)] \tag{43}$$

Here, $f(\cdot) := h(\mathcal{A}(\cdot))$ where $\mathcal{A}$ is the forward operator and $h(\boldsymbol{x})$ is the multivariate normal distribution with mean $\boldsymbol{y}$ and the covariance $\sigma^2 \boldsymbol{I}$. Therefore, we have

$$J(f, p(\boldsymbol{x}_0|\boldsymbol{x}_t)) = |\mathbb{E}[f(\boldsymbol{x}_0)] - f(\mathbb{E}[\boldsymbol{x}_0])| = |\mathbb{E}[f(\boldsymbol{x}_0)] - f(\hat{\boldsymbol{x}}_0)| \tag{44}$$

$$= |\mathbb{E}[h(\mathcal{A}(\boldsymbol{x}_0))] - h(\mathcal{A}(\hat{\boldsymbol{x}}_0))| \tag{45}$$

$$\leq \int |h(\mathcal{A}(\boldsymbol{x}_0)) - h(\mathcal{A}(\hat{\boldsymbol{x}}_0))| dP(\boldsymbol{x}_0|\boldsymbol{x}_t) \tag{46}$$

$$\overset{(b)}{\leq} \frac{d}{\sqrt{2\pi\sigma^2}} e^{-1/2\sigma^2} \int \|\mathcal{A}(\boldsymbol{x}_0) - \mathcal{A}(\hat{\boldsymbol{x}}_0)\| dP(\boldsymbol{x}_0|\boldsymbol{x}_t) \tag{47}$$

$$\overset{(c)}{\leq} \frac{d}{\sqrt{2\pi\sigma^2}} e^{-1/2\sigma^2} \|\nabla_{\boldsymbol{x}} \mathcal{A}(\boldsymbol{x})\| \int \|\boldsymbol{x}_0 - \hat{\boldsymbol{x}}_0\| dP(\boldsymbol{x}_0|\boldsymbol{x}_t) \tag{48}$$

$$\overset{(d)}{\leq} \frac{d}{\sqrt{2\pi\sigma^2}} e^{-1/2\sigma^2} \|\nabla_{\boldsymbol{x}} \mathcal{A}(\boldsymbol{x})\| m_1 \tag{49}$$

where $dP(\boldsymbol{x}_0|\boldsymbol{x}_t) = p(\boldsymbol{x}_0|\boldsymbol{x}_t) \, d\boldsymbol{x}_0$, (b) is the result of Lemma 3, (c) is from the intermediate value theorem, and (d) is from Proposition 2. $\qquad\square$

## B  INVERSE PROBLEM SETUP

**Super-resolution.** The forward model for super-resolution is defined as

$$\boldsymbol{y} \sim \mathcal{N}(\boldsymbol{y}|\boldsymbol{L}^f \boldsymbol{x}, \sigma^2 \boldsymbol{I}), \qquad \text{(Gaussian)} \tag{50}$$

$$\boldsymbol{y} \sim \mathcal{P}(\boldsymbol{y}|\boldsymbol{L}^f \boldsymbol{x}; \lambda), \qquad \text{(Poisson)} \tag{51}$$

where $\boldsymbol{L}^f \in \mathbb{R}^{n \times d}$ represents the bicubic downsampling block Hankel matrix with the factor $f$, and $\mathcal{P}$ denotes the Poisson distribution with the parameter $\lambda$.

**Inpainting.** For both box-type and random-type inpainting, the forward model reads

$$\boldsymbol{y} \sim \mathcal{N}(\boldsymbol{y}|\boldsymbol{P}\boldsymbol{x}, \sigma^2 \boldsymbol{I}), \qquad \text{(Gaussian)} \tag{52}$$

$$\boldsymbol{y} \sim \mathcal{P}(\boldsymbol{y}|\boldsymbol{P}\boldsymbol{x}; \lambda), \qquad \text{(Poisson)} \tag{53}$$

where $\boldsymbol{P} \in \{0, 1\}^{n \times d}$ is the masking matrix that consists of elementary unit vectors.

**Linear Deblurring.** For both Gaussian and motion deblurring, the measurement model is given as

$$\boldsymbol{y} \sim \mathcal{N}(\boldsymbol{y}|\boldsymbol{C}^\psi \boldsymbol{x}, \sigma^2 \boldsymbol{I}), \qquad \text{(Gaussian)} \tag{54}$$

$$\boldsymbol{y} \sim \mathcal{P}(\boldsymbol{y}|\boldsymbol{C}^\psi \boldsymbol{x}; \lambda), \qquad \text{(Poisson)} \tag{55}$$

where $\boldsymbol{C}^\psi \in \mathbb{R}^{n \times d}$ is the block Hankel matrix that effectively induces convolution with the given blur kernel $\psi$.

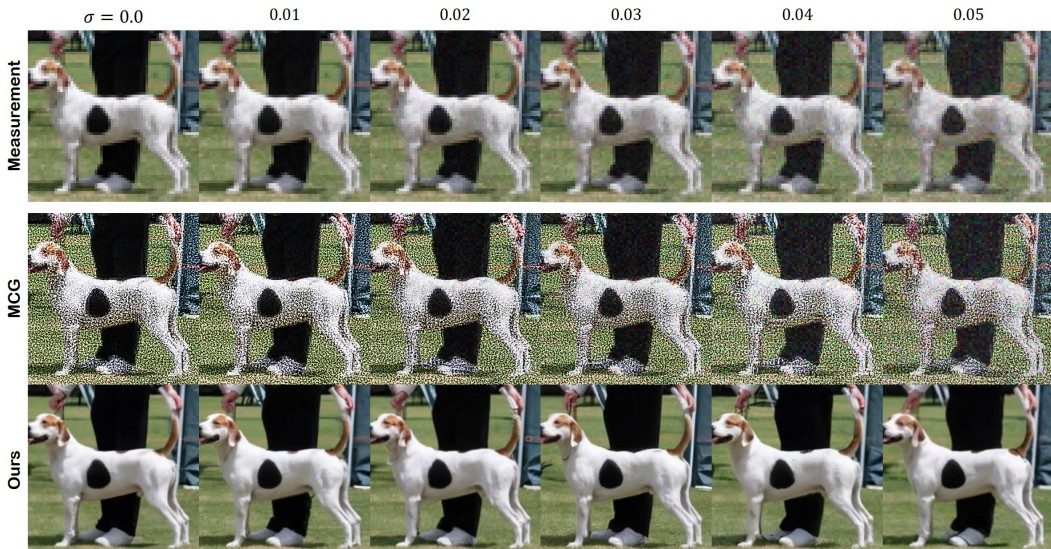

Figure 7: Failure cases of MCG (Chung et al., 2022a) on noisy inverse problems due to noise amplification.

**Nonlinear deblurring.** We leverage the nonlinear blurring process that was proposed in the GOPRO dataset (Nah et al., 2017), where the blurring process is not defined as a convolution, but rather as an integration of sharp images through the time frame. Specifically, in the discrete sense the measurement model reads

$$\boldsymbol{y} = \int b \left( \frac{1}{M} \sum_{i=1}^{M} \boldsymbol{x}[i] \right), \quad i = 1, \ldots, T, \tag{56}$$

where $b(\boldsymbol{x}) = \boldsymbol{x}^{1/2.2}$ is the nonlinear camera response function, and $T$ denotes the total time frames. While we could directly use (56) as our forward model, note that this is only possible when we have multiple sharp time frames at hand (e.g. when leveraging GOPRO dataset directly). Recently, there was an effort to *distill* the forward model through a neural network (Tran et al., 2021). Particularly, when we have a set of blurry-sharp image pairs $\{(\boldsymbol{x}_i, \boldsymbol{y}_i)\}_{i=1}^{N}$, one can train a neural network to estimate the forward model as

$$\phi^* = \arg\min_{\theta} \sum_{i=1}^{N} \|\boldsymbol{y}_i - \mathcal{F}_{\phi}(\boldsymbol{x}_i, \mathcal{G}_{\phi}(\boldsymbol{x}_i, \boldsymbol{y}_i))\|, \tag{57}$$

where $\mathcal{G}_{\phi}(\boldsymbol{x}_i, \boldsymbol{y}_i)$ extracts the implicit kernel information from the pair, and $\mathcal{F}_{\phi}$ takes in $\boldsymbol{x}_i, \mathcal{G}_{\phi}(\boldsymbol{x}_i, \boldsymbol{y}_i)$ to generate the blurry image. When using $\mathcal{F}_{\phi}$ at deployment to generate new synthetic blurry images, one can simply replace $\mathcal{G}_{\phi}(\boldsymbol{x}_i, \boldsymbol{y}_i)$ with a Gaussian random vector $\boldsymbol{k}$. Consequently, our forward model reads

$$\boldsymbol{y} \sim \mathcal{N}(\boldsymbol{y}|\mathcal{F}_{\phi}(\boldsymbol{x}, \boldsymbol{k}), \sigma^2 \boldsymbol{I}), \ \boldsymbol{k} \in \mathbb{R}^k, \ \boldsymbol{k} \in \mathcal{N}(0, \sigma_k^2 I), \qquad \text{(Gaussian)} \tag{58}$$

$$\boldsymbol{y} \sim \mathcal{P}(\boldsymbol{y}|\mathcal{F}_{\phi}(\boldsymbol{x}, \boldsymbol{k}); \lambda), \ \boldsymbol{k} \in \mathbb{R}^k, \ \boldsymbol{k} \in \mathcal{N}(0, \sigma_k^2 I), \qquad \text{(Poisson)} \tag{59}$$

where $k$ is the dimensionality of the latent vector $\boldsymbol{k}$, and $\sigma_k^2$ is the variance of the vector.

**Phase Retrieval.** The forward measurement model is usually given as

$$\boldsymbol{y} \sim \mathcal{N}(\boldsymbol{y}||\boldsymbol{F}\boldsymbol{x}_0|, \sigma^2 \boldsymbol{I}), \qquad \text{(Gaussian)} \tag{60}$$

$$\boldsymbol{y} \sim \mathcal{P}(\boldsymbol{y}||\boldsymbol{F}\boldsymbol{x}_0|; \lambda), \qquad \text{(Poisson)} \tag{61}$$

where $\boldsymbol{F}$ denotes the 2D Discrete Fourier Trasform (DFT) matrix. In another words, the phase of the Fourier measurements are nulled, and our aim is to impute the missing phase information. As the problem is highly ill-posed, one typically incorporates the oversampling in order to induce the

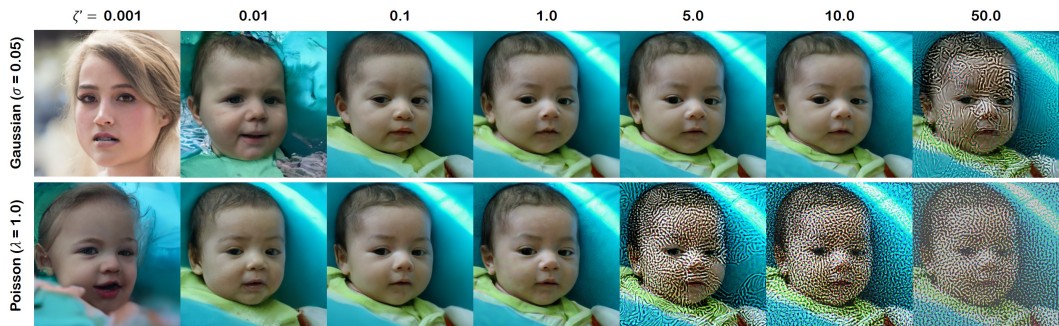

Figure 8: Effect of step size $\zeta'$ on the results

uniqueness condition (Hayes, 1982; Bruck & Sodin, 1979), usually specified as

$$\boldsymbol{y} \sim \mathcal{N}(\boldsymbol{y} \| \boldsymbol{FPx}_0 |, \sigma^2 \boldsymbol{I}), \qquad \text{(Gaussian)} \tag{62}$$
$$\boldsymbol{y} \sim \mathcal{P}(\boldsymbol{y} \| \boldsymbol{FPx}_0 |; \lambda), \qquad \text{(Poisson)} \tag{63}$$

where $\boldsymbol{P}$ denotes the oversampling matrix with ratio $k/n$.

**Poisson noise simulation.** To simulate the Poisson noise, we assume that each measurement pixel is a source of photon, where the number of photons is proportional to the discrete pixel value between 0 and 255. Thus, we sample noisy measurement values from the Poisson distribution with the mean value of the clean measurement values. Here, the clean measurement is $\mathcal{A}(\boldsymbol{x}_0)$, which is an image after applying the forward operation. Then, we clip the values by [0, 255] and normalize to [-1, 1].

## C ABLATION STUDIES AND DISCUSSION

### C.1 NOISE AMPLIFICATION BY PROJECTION

As discussed in the experiments, methods that rely on projections fail dramatically when solving inverse problems with excessive amount of noise in the measurement. Even worse, for many problems such as SR or deblurring, noise gets *amplified* during the projection step due to the operator transpose $\boldsymbol{A}^T$ being applied. This downside is clearly depicted in Fig. 7, where we show the failure cases of MCG (Chung et al., 2022a) on noisy super-resolution. In contrast, our method does not rely on such projections, and thus is much more robust to the corrupted measurements. Notably, we find that MCG also fails dramatically in SR even when there is no noise existent, while it performs well on some of the other tasks (e.g. inpainting). We can conclude that the proposed method works generally well across a broader range of inverse problems, whether or not there is noise in the measurement.

### C.2 EFFECT OF STEP SIZE $\zeta'$

There is one hyper-parameter in our DPS solver, and that is the step size. As this value is essentially the weight that is given to the likelihood (i.e. data consistency) of the inverse problem, we can expect that the values being too high or too low will cause problems. In Fig. 8, we show the trend of the reconstruction results when varying the step size $\zeta_i$. Note that we instead use the notation $\zeta' \triangleq \zeta_i \|\boldsymbol{y} - \mathcal{A}(\hat{\boldsymbol{x}}_0(\boldsymbol{x}_i))\|$ for brevity. Here, we see that with low values of $\zeta' < 0.1$, we achieve results that are not consistent with the given measurement. On the other hand, when we crank up the values too high ($\zeta' > 5$), we observe saturation arfiacts that tend to amplify the noise. From our experiments, we conclude that it is best practice to set the $\zeta'$ values in the range $[0.1, 1.0]$ for best results. Specific values for all the experiments are presented in Appendix D.

### C.3 OTHER STEP SIZE SCHEDULES

While the proposed step size schedule in C.2 yields good results, there can be many other choices that one can take. In this section, we conduct an ablation study to compare against other choices. Specifically, we test 100 images for Gaussian deblurring (Gaussian noise, $\sigma = 0.05$) on FFHQ, and

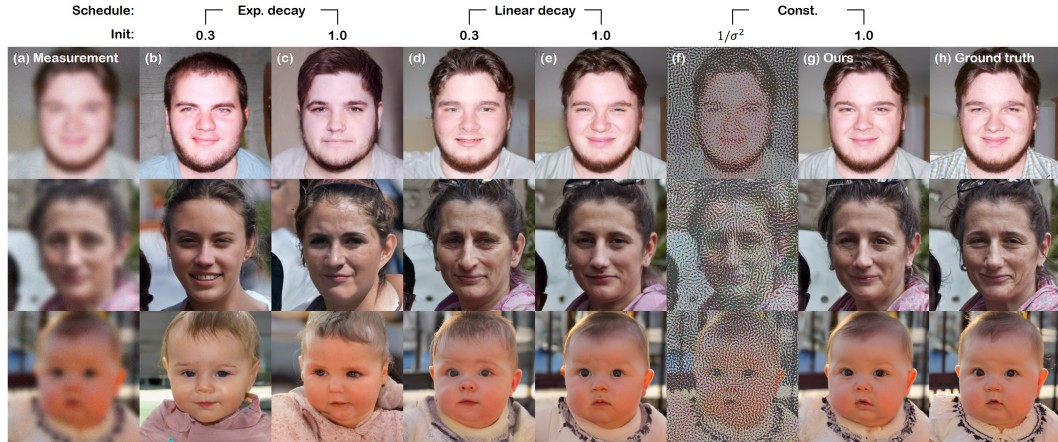

Figure 9: Ablation study on the choice of step size schedule for DPS. (a) Measurement, (b-c) exponential decay with initial values 0.3, 1.0, (d-e) linear decay with initial values 0.3, 1.0, (f) $\propto 1/\sigma^2$ (g) **ours**, (h) ground truth.

| Strategy | **Constant** | | **Linear decay** | | **Exponential decay** | |
|---|---|---|---|---|---|---|
| Initial value | 1.0 (**Ours**) | $1/\sigma^2$ | 0.3 | 1.0 | 0.3 | 1.0 |
| LPIPS $\downarrow$ | **0.247** $\pm$ 0.045 | 0.727 $\pm$ 0.038 | 0.287 $\pm$ 0.045 | 0.251 $\pm$ 0.044 | 0.421 $\pm$ 0.065 | 0.442 $\pm$ 0.108 |

Table 5: Ablation study on step size scheduling. **Bold**: best, underline: second best.

compute the average perceptual distance (LPIPS) against the ground truth. We compare against the following three choices: 1) Linearly decaying steps $\zeta'_i = \zeta'_{rminit} \times \left(1 - \frac{i}{N}\right)$, 2) exponentially decaying steps $\zeta'_i = \zeta'_{\text{init}} \times \gamma^i$, with $\gamma = 0.99$, 3) directly using step size proportional to $1/\sigma^2$ as in eq. 16.

We present qualitative analysis in Fig. 9. From the figure, it is clear that the proposed schedule produces the best result that most closely matches the ground truth in terms of perception. For decaying step sizes, we often yield results that are coarsely similar to the ground truth, but varies in the fine details, as the information about the measurement is less incorporated in the later steps of the diffusion. From Fig. 9, we see that taking step sizes proportional to $1/\sigma^2$, motivated by direct derivation from the gaussian forward model, yields poor results. We see similar results with the quantitative metrics presented in Table. 5.

### C.4 POISSON INVERSE PROBLEMS

For inverse problems corrupted with Poisson noise, more care needs to be taken compared to the Gaussian noise counterparts, as the noise is signal-dependent and therefore harder to account for. In this section, we discuss the different choices of likelihood functions that can be made, and clarify the choice (20) used in all our experiments. One straightforward option is to directly use the Poisson likelihood model without the Gaussian approximation. From (17), we have that

$$\log p(\boldsymbol{y}|\boldsymbol{x}_0) = \sum_{j=1}^{n} \log[\mathcal{A}(\boldsymbol{x}_0)]_j - [\mathcal{A}(\boldsymbol{x}_0)]_j - \log(\boldsymbol{y}_j!) \tag{64}$$

$$\nabla_{\boldsymbol{x}_t} \log p(\boldsymbol{y}|\boldsymbol{x}_0) = -\alpha \nabla_{\boldsymbol{x}_t} \left[\sum_{j=1}^{n} \log[\mathcal{A}(\boldsymbol{x}_0)]_j - [\mathcal{A}(\boldsymbol{x}_0)]_j\right], \tag{65}$$

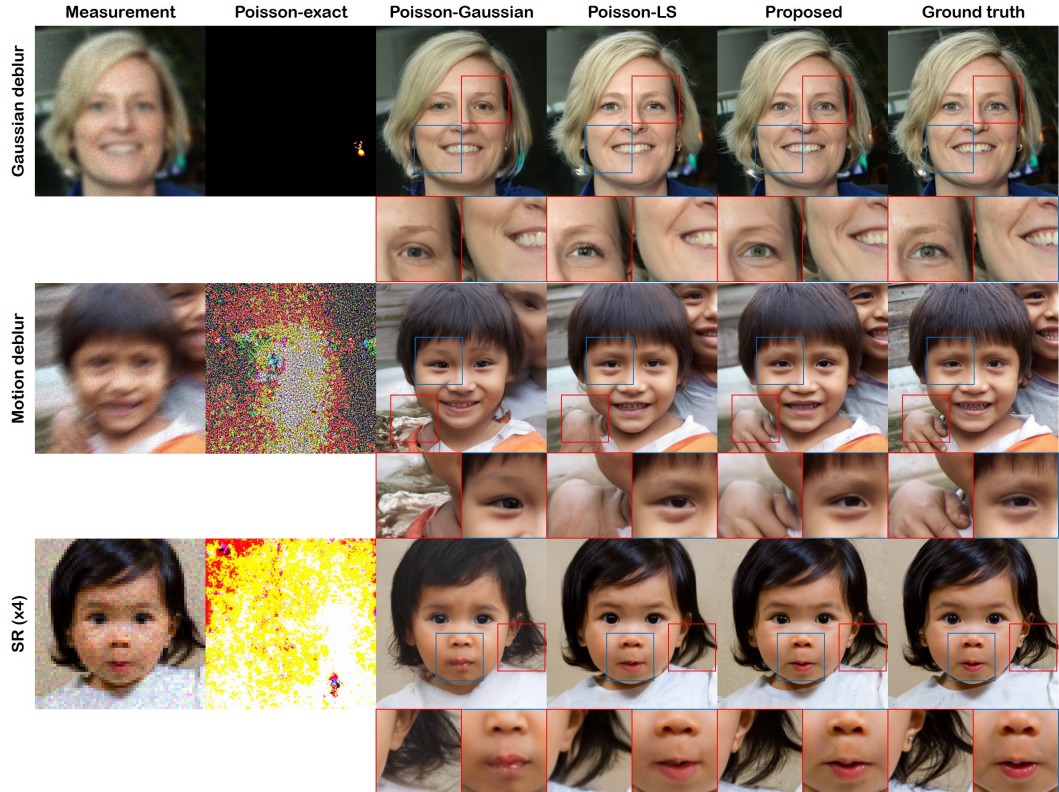

Figure 10: Differences in the reconstruction results when using different choices for imposing data consistency for Poisson linear inverse problems.

which we refer to as Poisson-direct. Moreover, one can use the Gaussian approximated version of the Poisson measurement model given in (18)

$$\log p(\boldsymbol{y}|\boldsymbol{x}_0) = \sum_{j=1}^{n} -\frac{1}{2} \log \left[2\pi[\mathcal{A}(\boldsymbol{x}_0)]_j\right] - \frac{(\boldsymbol{y}_j - [\mathcal{A}(\boldsymbol{x}_0)]_j)^2}{2[\mathcal{A}(\boldsymbol{x}_0)]_j} \tag{66}$$

$$\nabla_{\boldsymbol{x}_t} \log p(\boldsymbol{y}|\boldsymbol{x}_0) = \alpha \nabla_{\boldsymbol{x}_t} \left[\sum_{j=1}^{n} \frac{1}{2} \log \left[2\pi[\mathcal{A}(\boldsymbol{x}_0)]_j\right] + \frac{(\boldsymbol{y}_j - [\mathcal{A}(\boldsymbol{x}_0)]_j)^2}{2[\mathcal{A}(\boldsymbol{x}_0)]_j}\right], \tag{67}$$

which we refer to as Poisson-Gaussian. Next, we can use our choice in (19) to arrive at (20), which is the proposed method. Finally, while irrelevant with the noise model, we can also still use the same least sqaures (LS) method used for Gaussian noise (we refer to this method as Poisson-LS), as due to the central limit theorem, Poisson noise is nearly Gaussian in the high SNR level regime. In Fig. 10, we show representative results achieved by using each choice. From the experiments, we observe that Poisson-direct is unstable due to the log term in the likelihood, hence often diverging. We also observe that the residual $\boldsymbol{y} - \mathcal{A}(\hat{\boldsymbol{x}}_0)$ fails to converge, hinting that the information from the measurement is not effectively integrated into the generative process. For Poisson-Gaussian, we see that the weighting term of the MSE is problematic, and this term prevents the process from proper convergence. Both the proposed method and Poisson-LS are stable, but Poisson-LS tends to blur out the relevant details from the reconstruction, while Poisson-shot preserves the high-frequency details better, and does not alter the identity of the ground truth person.

## C.5 SAMPLING SPEED

As widely known in the literature, diffusion model-based methods are heavily dependent on the number of neural function evaluations (NFE). We investigate the performance in terms of LPIPS

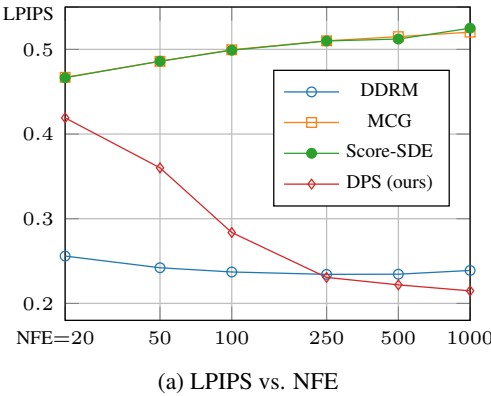

| Method | Wall-clock time [s] |
|---|---|
| Score-SDE (Song et al., 2021b) | 36.71 |
| DDRM (Kawar et al., 2022) | 2.029 |
| MCG (Chung et al., 2022a) | 80.10 |
| PnP-ADMM (Chan et al., 2016) | 3.631 |
| BKS-styleGAN2 (Tran et al., 2021) | 891.8 |
| BKS-generic (Tran et al., 2021) | 93.23 |
| ER (Fienup, 1982) | 5.604 |
| HIO (Fienup & Dainty, 1987) | 6.317 |
| OSS (Rodriguez et al., 2013) | 15.65 |
| **Ours** | **78.52** |

(a) LPIPS vs. NFE

(b) Runtime for each algorithm in Wall-clock time: Computed with a single GTX 2080Ti GPU.

Figure 11: Ablation studies performed with SR×4 task on FFHQ 256×256 data, and the runtime analysis of the different algorithms.

with respect to the change in NFEs in Fig. 11a. For the experiment, we take the case of noisy SR×4, which is a problem where DDRM tends to perform well, in contrast to other problems, e.g. inpainting. In the high NFE regime ($\geq 250$), DPS outperforms all the other methods, whereas in the low NFE regime ($\leq 100$), DDRM takes over. This can be attributed to DDIM (Song et al., 2021a) sampling strategy that DDRM adopts, known for better performance in the low NFE regimes. Orthogonal to the direction presented in this work, devising a method to improve the performance of DPS in such regime with advanced samplers (e.g. Lu et al. (2022); Liu et al. (2022)) would benefit the method.

## C.6 LIMITATIONS

Inheriting the characteristics of the diffusion model-based methods, the proposed method is relatively slow, as can be seen in the runtime analysis of Fig. 11b. However, we note that our method is still faster than the GAN-based optimization methods, as we do not have to finetune the network itself. Moreover, the slow sampling speed could be mitigated with the incorporation of advanced samplers. Our method tends to preserve the high frequency details (e.g. beard, hair, texture) of the image, while methods such as DDRM tends to produce rather blurry images. In the qualitative view, and in the perception oriented metris (i.e. FID, LPIPS), our method clearly outperforms DDRM. In contrast, in standard distortion metrics such as PSNR, our method underperforms DDRM. This can be explained by the perception-distortion tradeoff phenomena (Blau & Michaeli, 2018), where preserving high frequency details may actually *penalize* the reconstructions from having better distortion metrics. Finally, we note that the reconstruction quality of phase retrieval is not as robust as compared to other problems - linear inverse problems and nonlinear deblurring. Due to the stochasticity, we often encounter failures among the posterior samples, which can be potentially counteracted by simply taking multiple samples, as was done in other methods. Devising methods to stabilize the samplers, especially for nonlinear phase retrieval problems, would be a promising direction of research.

## D EXPERIMENTAL DETAILS

### D.1 IMPLEMENTATION DETAILS

**Step size.** Here, we list the step sizes used in our DPS algorithm for each problem setting.

- Linear inverse problem
    - Gaussian measurement noise
        * FFHQ
            · Super-resolution: $\zeta_i = 1/\|\boldsymbol{y} - \mathcal{A}(\hat{\boldsymbol{x}}_0(\boldsymbol{x}_i))\|$
            · Inpainting: $\zeta_i = 1/\|\boldsymbol{y} - \mathcal{A}(\hat{\boldsymbol{x}}_0(\boldsymbol{x}_i))\|$
            · Deblurring (Gauss): $\zeta_i = 1/\|\boldsymbol{y} - \mathcal{A}(\hat{\boldsymbol{x}}_0(\boldsymbol{x}_i))\|$
            · Deblurring (motion): $\zeta_i = 1/\|\boldsymbol{y} - \mathcal{A}(\hat{\boldsymbol{x}}_0(\boldsymbol{x}_i))\|$

* ImageNet
  · Super-resolution: $\zeta_i = 1/\|\boldsymbol{y} - \mathcal{A}(\hat{\boldsymbol{x}}_0(\boldsymbol{x}_i))\|$
  · Inpainting: $\zeta_i = 1/\|\boldsymbol{y} - \mathcal{A}(\hat{\boldsymbol{x}}_0(\boldsymbol{x}_i))\|$
  · Deblurring (Gauss): $\zeta_i = 0.4/\|\boldsymbol{y} - \mathcal{A}(\hat{\boldsymbol{x}}_0(\boldsymbol{x}_i))\|$
  · Deblurring (motion): $\zeta_i = 0.6/\|\boldsymbol{y} - \mathcal{A}(\hat{\boldsymbol{x}}_0(\boldsymbol{x}_i))\|$
– Poisson measurement noise
  * FFHQ
    · Super-resolution: $\zeta_i = 0.3/\|\boldsymbol{y} - \mathcal{A}(\hat{\boldsymbol{x}}_0(\boldsymbol{x}_i))\|$
    · Deblurring (Gauss): $\zeta_i = 0.3/\|\boldsymbol{y} - \mathcal{A}(\hat{\boldsymbol{x}}_0(\boldsymbol{x}_i))\|$
    · Deblurring (motion): $\zeta_i = 0.3/\|\boldsymbol{y} - \mathcal{A}(\hat{\boldsymbol{x}}_0(\boldsymbol{x}_i))\|$
- Nonlinear inverse problem
  – Gaussian measurement noise
    * FFHQ
      · Phase retrieval: $\zeta_i = 0.4/\|\boldsymbol{y} - \mathcal{A}(\hat{\boldsymbol{x}}_0(\boldsymbol{x}_i))\|$
      · non-uniform deblurring: $\zeta_i = 1.0/\|\boldsymbol{y} - \mathcal{A}(\hat{\boldsymbol{x}}_0(\boldsymbol{x}_i))\|$

**Score functions used.** Pre-trained score function for the FFHQ dataset was taken from Choi et al. (2021)[7], and the score function for the ImageNet dataset was taken from Dhariwal & Nichol (2021)[8].

**Compute time.** All experiments were performed on a single RTX 2080Ti GPU. FFHQ experiments take about 95 seconds per image (1000 NFE), while ImageNet experiments take about 600 seconds per image (1000 NFE) for reconstruction due to the much larger network size.

**Code availability.** Code is available at `https://github.com/DPS2022/diffusion-posterior-sampling`.

## D.2 COMPARISON METHODS

For DDRM, MCG, Score-SDE, and our method we use the same checkpoint for the score functions.

**DDRM.** All experiments were performed with the default setting of $\eta_B = 1.0, \eta = 0.85$, and leveraging DDIM (Song et al., 2021a) sampling for 20 NFEs. For the Gaussian deblurring experiment, the forward model was implemented by separable 1D convolutions for efficient SVD.

**MCG.** We set the same values of $\alpha$ that are used in our methods (DPS). At each step, the additional data consistency steps are applied as Euclidean projections onto the measurement set $\mathcal{C} := \{\boldsymbol{x}_i | \mathcal{A}(\boldsymbol{x}_i) = \boldsymbol{y}_i, \; \boldsymbol{y}_i \sim p(\boldsymbol{y}_i | \boldsymbol{y}_0)\}$.

**Score-SDE.** Score-SDE solves inverse problems by iteratively applying denoising followed by data consistency projections. As in MCG, we apply Euclidean projections onto the measurment set $\mathcal{C}$.

**PnP-ADMM.** We take the implementation from the `scico` library (Balke et al., 2022). The parameters are set as follows: $\rho = 0.2$ (ADMM penalty parameter), `maxiter`$= 12$. For proximal mappings, we utilize the pretrained DnCNN Zhang et al. (2017) denoiser.

**ADMM-TV.** We minimize the following objective

$$\min_{\boldsymbol{x}} \frac{1}{2}\|\boldsymbol{y} - \mathcal{A}(\boldsymbol{x}_0)\|_2^2 + \lambda\|\boldsymbol{D}\boldsymbol{x}_0\|_{2,1}, \tag{68}$$

where $\boldsymbol{D} = [\boldsymbol{D}_x, \boldsymbol{D}_y]$ computes the finite difference with respect to both axes, $\lambda$ is the regularization weight, and $\|\cdot\|_{2,1}$ implements the isotropic TV regularization. Note that the optimization is solved with ADMM, and hence we have an additional parameter $\rho$. We take the implementation from the `scico` library (Balke et al., 2022). The parameters $\lambda, \rho$ were found with grid search for each optimization problems. We use the following settings: $(\lambda, \rho) = (2.7e - 2, 1.4e - 1)$ for deblurring, $(\lambda, \rho) = (2.7e - 2, 1.0e - 2)$ for SR and inpainting.

**ER, HIO, OSS.** For all algorithms, we initialize a real signal by sampling from the normal distribution as the problem statement of (Fienup, 1982). For the object domain constraint, we apply both the

[7]`https://github.com/jychoi118/ilvr_adm`
[8]`https://github.com/openai/guided-diffusion`. Unconditional version

non-negative constraint and the finite support constraint. We set the number of iterations to 10,000 for sufficient convergence. To mitigate the instability of reconstruction depending on initialization, we repeat each algorithm four times per data and report the best one with the smallest mean squared error between the measurement and amplitude of the estimation in the Fourier domain. In the case of HIO and OSS, we set $\beta$ to 0.9, which yields best results.

## E  FURTHER EXPERIMENTAL RESULTS

We first provide quantitative evaluations based on the standard PSNR and SSIM metrics in Table 6 and Table 7.

| Method | SR ($\times 4$) | | Inpaint (box) | | Inpaint (random) | | Deblur (gauss) | | Deblur (motion) | |
|---|---|---|---|---|---|---|---|---|---|---|
| | PSNR ↑ | SSIM ↑ | PSNR ↑ | SSIM ↑ | PSNR ↑ | SSIM ↑ | PSNR ↑ | SSIM ↑ | PSNR ↑ | SSIM ↑ |
| DPS (ours) | 25.67 | 0.852 | **22.47** | **0.873** | **25.23** | **0.851** | 24.25 | 0.811 | **24.92** | **0.859** |
| DDRM (Kawar et al., 2022) | 25.36 | 0.835 | 22.24 | 0.869 | 9.19 | 0.319 | 23.36 | 0.767 | - | - |
| MCG (Chung et al., 2022a) | 20.05 | 0.559 | 19.97 | 0.703 | 21.57 | 0.751 | 6.72 | 0.051 | 6.72 | 0.055 |
| PnP-ADMM (Chan et al., 2016) | **26.55** | **0.865** | 11.65 | 0.642 | 8.41 | 0.325 | **24.93** | **0.812** | 24.65 | 0.825 |
| Score-SDE (Song et al., 2021b) (ILVR (Choi et al., 2021)) | 17.62 | 0.617 | 18.51 | 0.678 | 13.52 | 0.437 | 7.12 | 0.109 | 6.58 | 0.102 |
| ADMM-TV | 23.86 | 0.803 | 17.81 | 0.814 | 22.03 | 0.784 | 22.37 | 0.801 | 21.36 | 0.758 |

Table 6: Quantitative evaluation (PSNR, SSIM) of solving linear inverse problems on FFHQ 256×256-1k validation dataset. **Bold**: best, underline: second best.

| Method | SR ($\times 4$) | | Inpaint (box) | | Inpaint (random) | | Deblur (gauss) | | Deblur (motion) | |
|---|---|---|---|---|---|---|---|---|---|---|
| | PSNR ↑ | SSIM ↑ | PSNR ↑ | SSIM ↑ | PSNR ↑ | SSIM ↑ | PSNR ↑ | SSIM ↑ | PSNR ↑ | SSIM ↑ |
| DPS (ours) | 23.87 | 0.781 | **18.90** | 0.794 | **22.20** | **0.739** | 21.97 | **0.706** | 20.55 | 0.634 |
| DDRM (Kawar et al., 2022) | **24.96** | **0.790** | 18.66 | **0.814** | 14.29 | 0.403 | **22.73** | 0.705 | - | - |
| MCG (Chung et al., 2022a) | 13.39 | 0.227 | 17.36 | 0.633 | 19.03 | 0.546 | 16.32 | 0.441 | 5.89 | 0.037 |
| PnP-ADMM (Chan et al., 2016) | 23.75 | 0.761 | 12.70 | 0.657 | 8.39 | 0.300 | 21.81 | 0.669 | **21.98** | **0.702** |
| Score-SDE (Song et al., 2021b) (ILVR (Choi et al., 2021)) | 12.25 | 0.256 | 16.48 | 0.612 | 18.62 | 0.517 | 15.97 | 0.436 | 7.21 | 0.120 |
| ADMM-TV | 22.17 | 0.679 | 17.96 | 0.785 | 20.96 | 0.676 | 19.99 | 0.634 | 20.79 | 0.677 |

Table 7: Quantitative evaluation (PSNR, SSIM) of solving linear inverse problems on ImageNet 256×256-1k validation dataset. **Bold**: best, underline: second best.

Further experimental results that show the ability of our method to sample multiple reconstructions are presented in Figs. 12,13, 14, 15, 16, 17 (Gaussian measurement with $\sigma = 0.05$), and Fig. 18,19 (Poisson measurement with $\lambda = 1.0$).

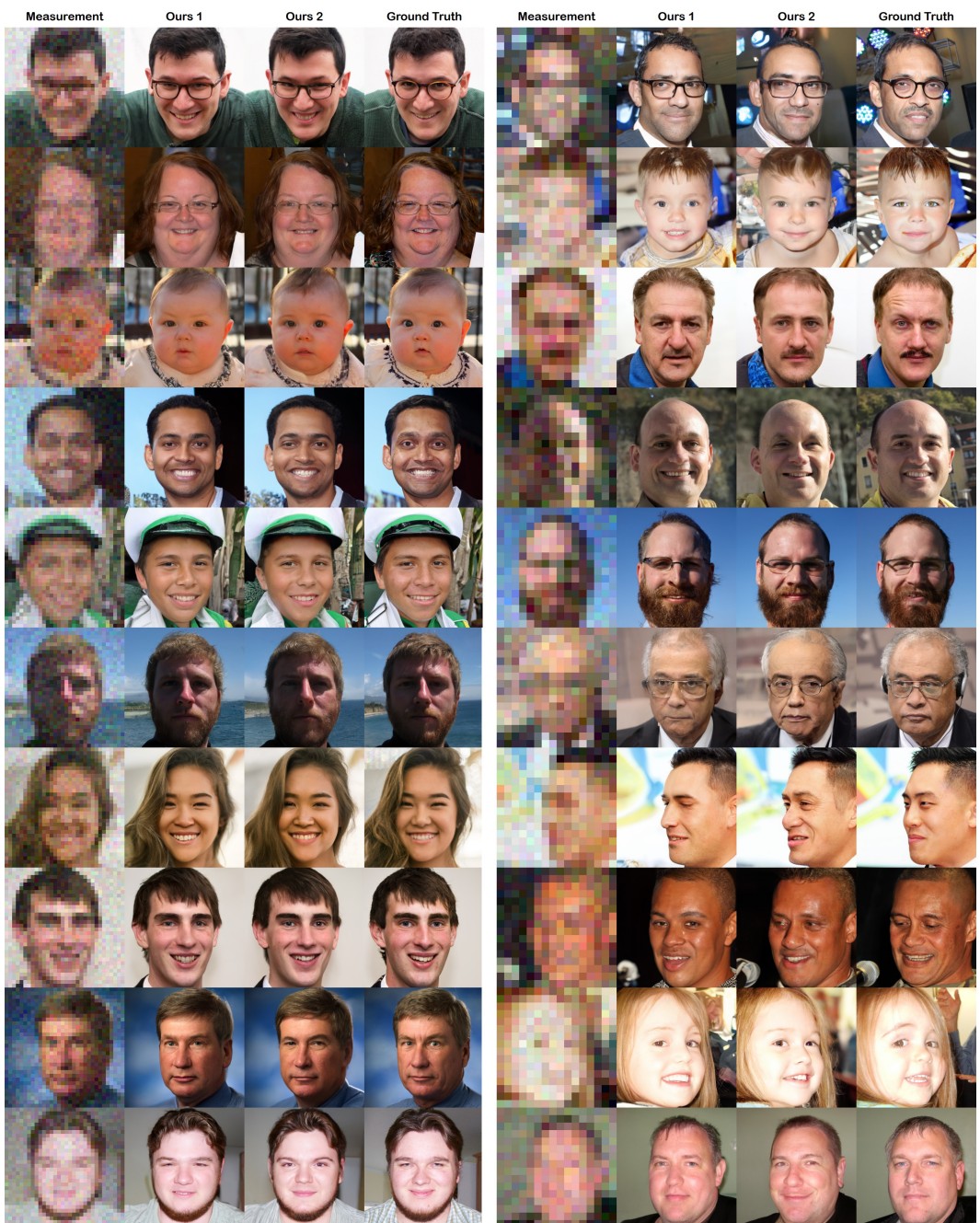

Figure 12: SR (Left ×8, Right ×16), results on the FFHQ (Karras et al., 2019) 256 × 256 dataset.

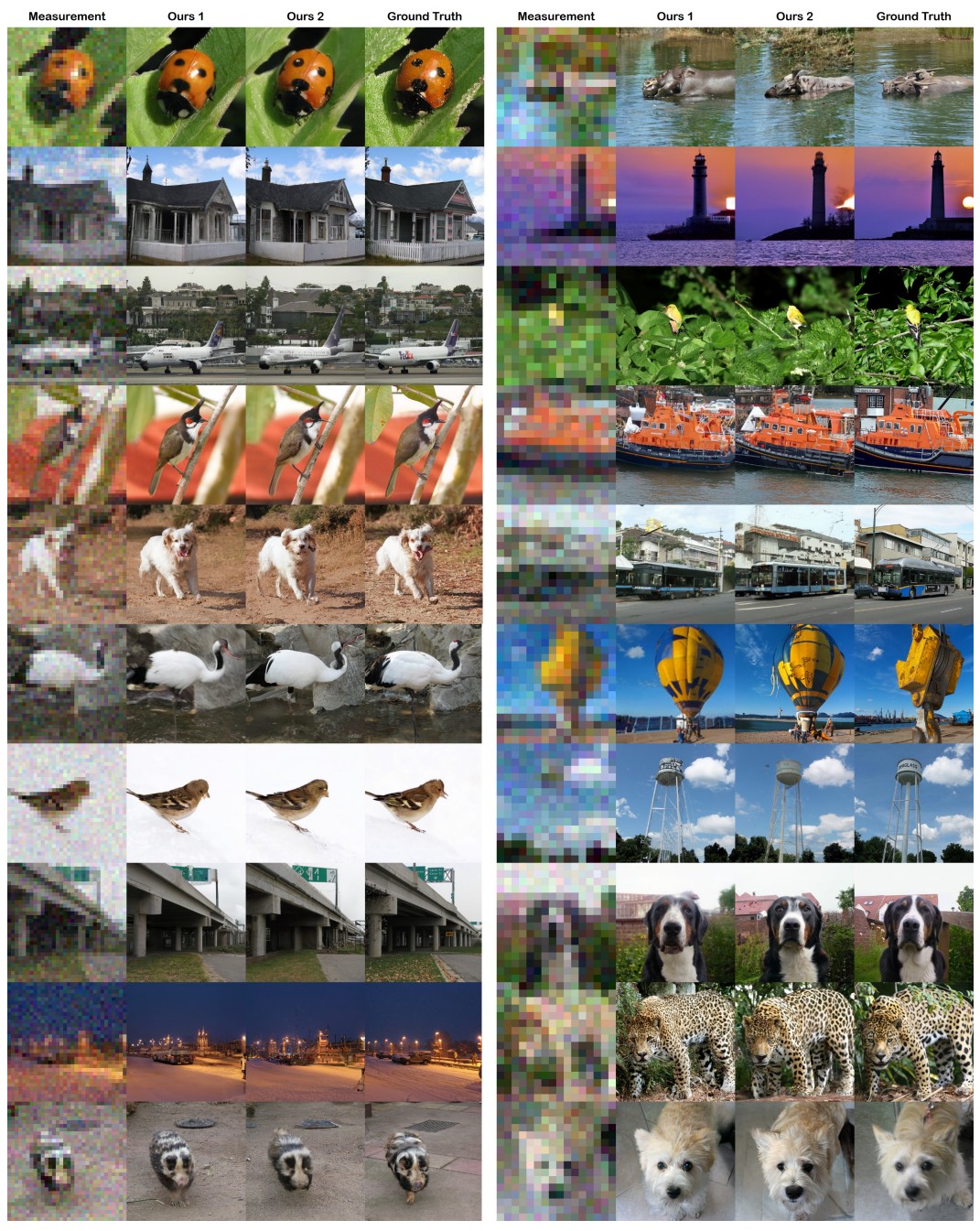

Figure 13: SR (Left ×8, Right ×16), results on the ImageNet (Deng et al., 2009) 256 × 256 dataset.

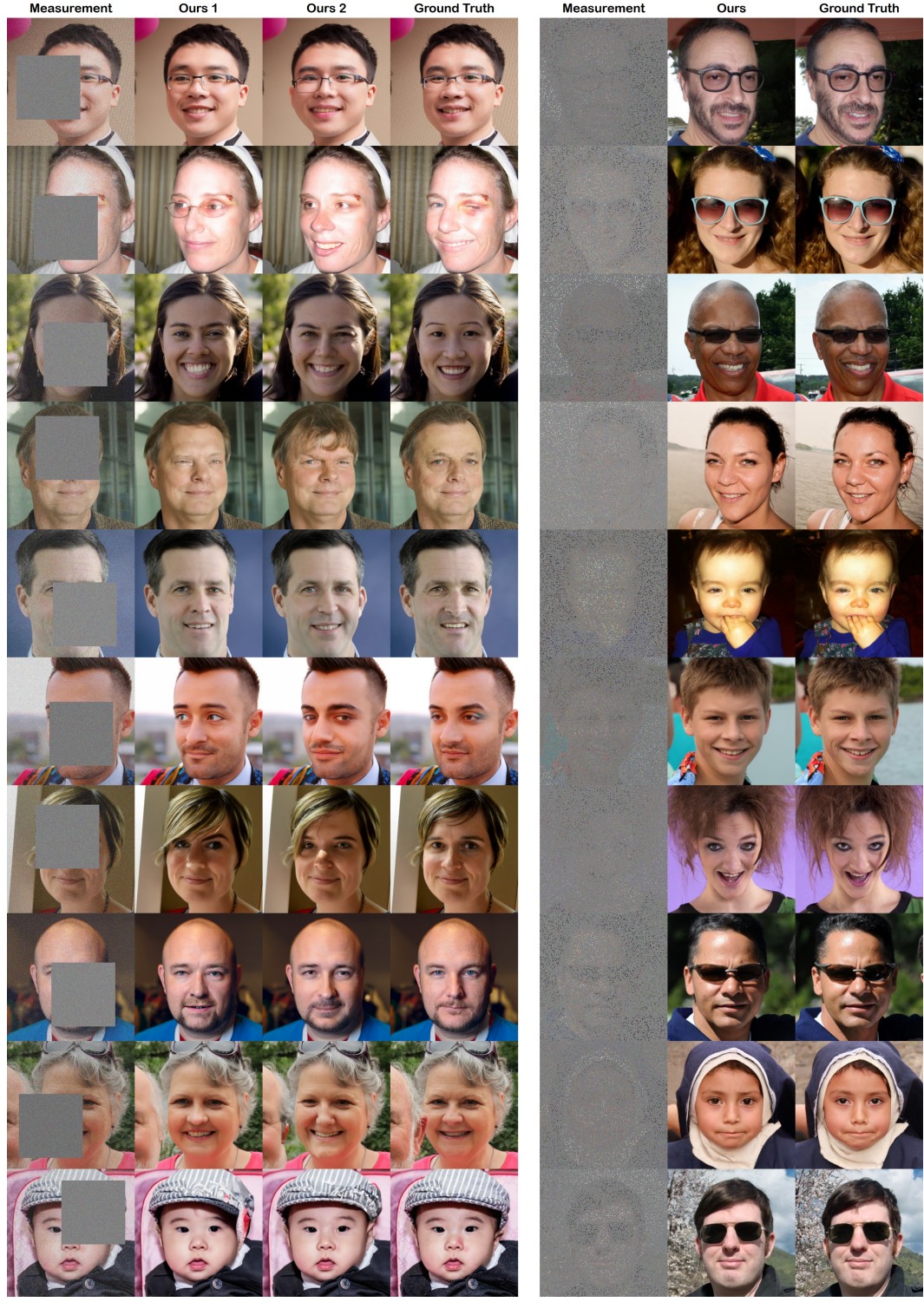

Figure 14: Inpainting results (Left 128×128 box, Right 92% random) on the FFHQ (Karras et al., 2019) 256 × 256 dataset.

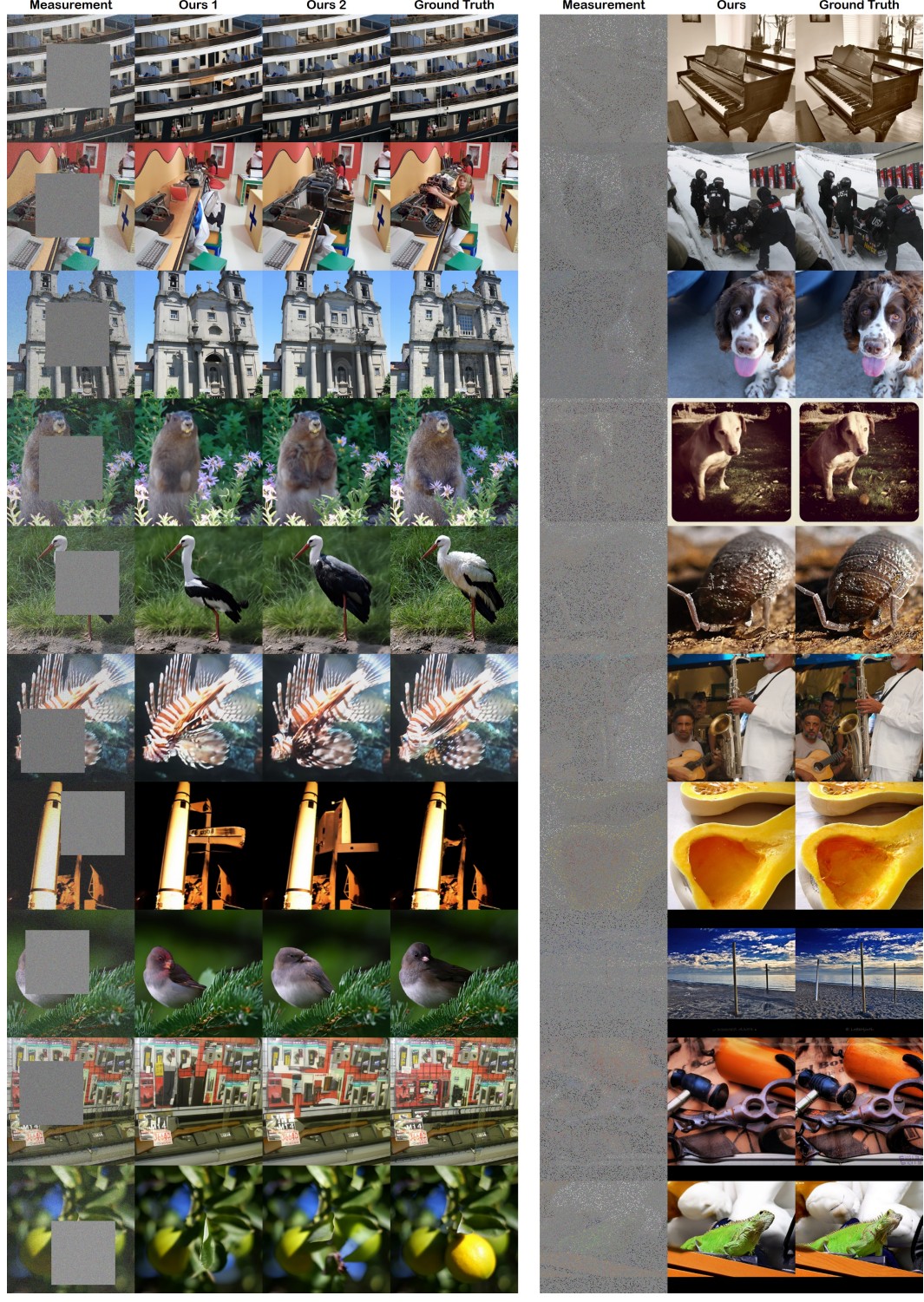

Figure 15: Inpainting results (Left 128×128 box, Right 92% random) on the ImageNet (Deng et al., 2009) 256 × 256 dataset.

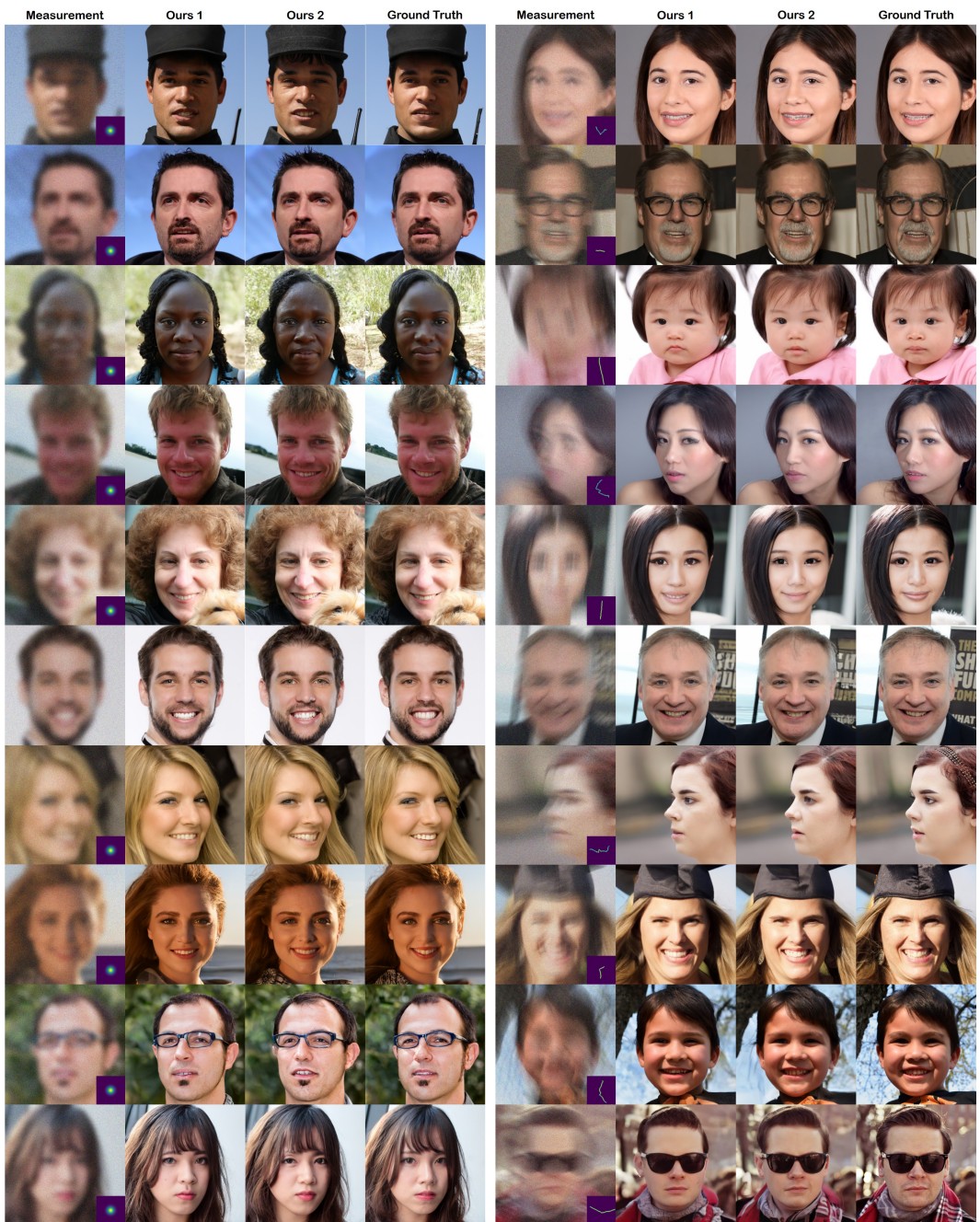

Figure 16: Deblurring results (Left Gaussian, Right motion) on the FFHQ (Karras et al., 2019) $256 \times 256$ dataset.

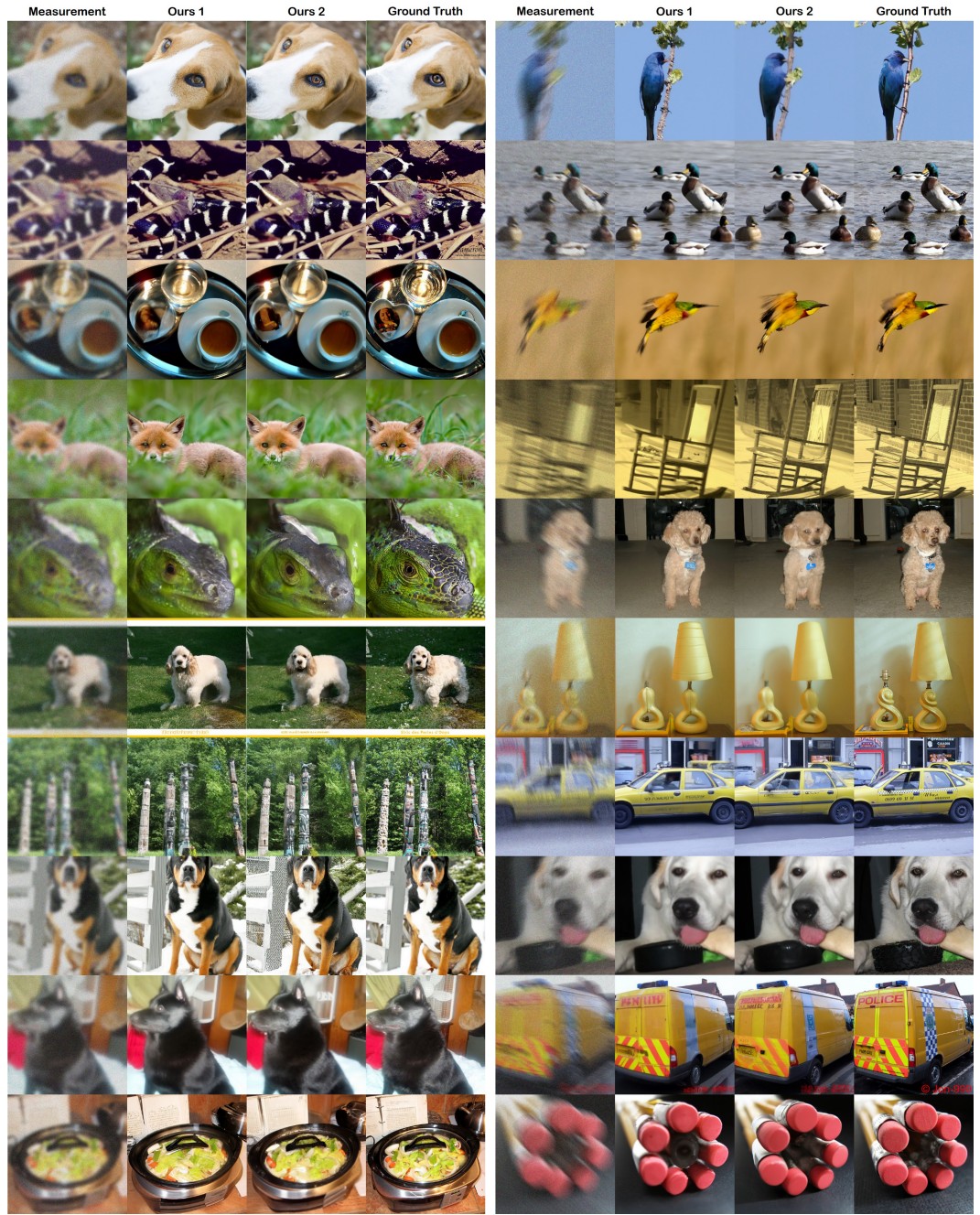

Figure 17: Deblurring results (Left Gaussian, Right motion) on the ImageNet (Karras et al., 2019) $256 \times 256$ dataset.

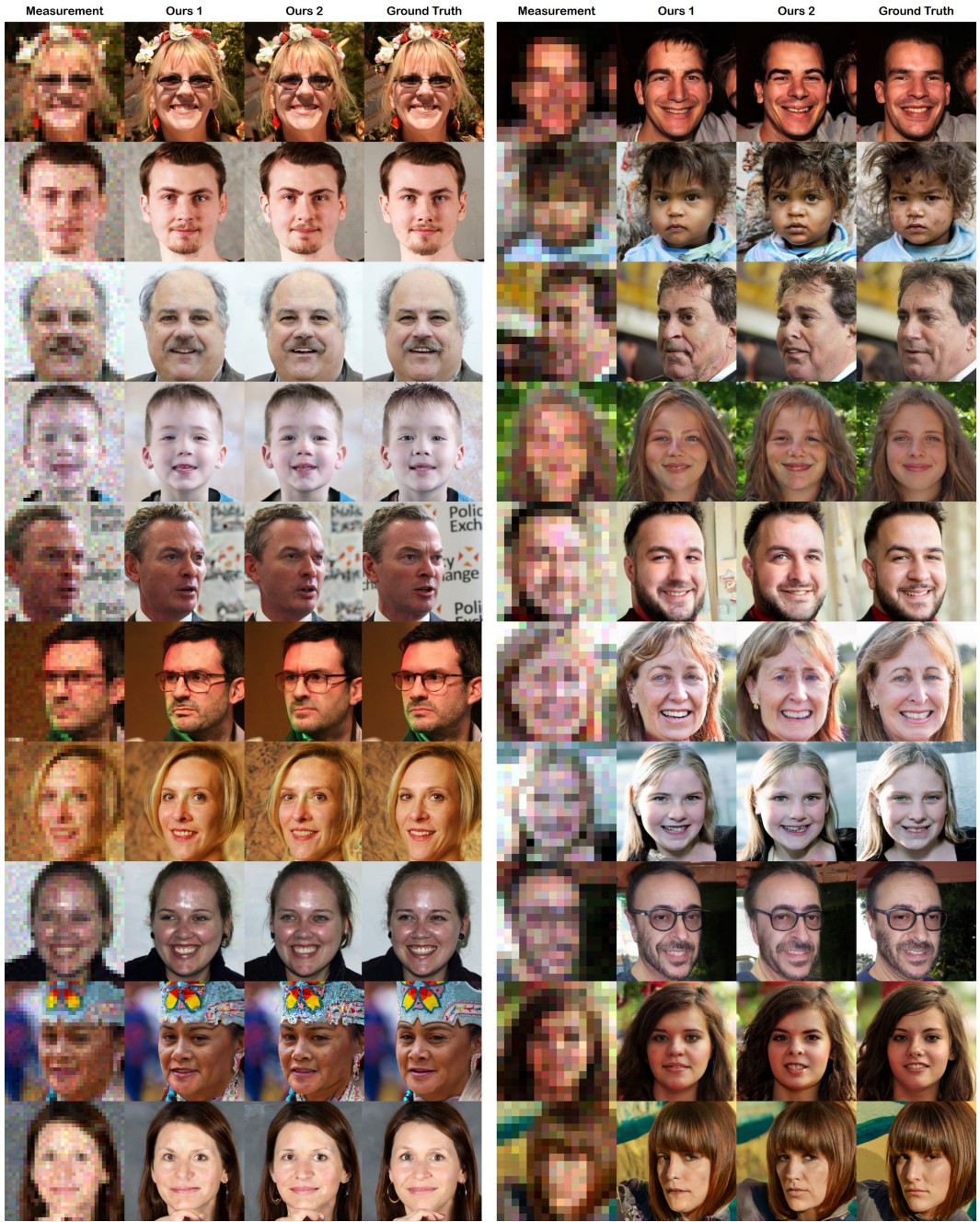

Figure 18: SR (Left ×8, Right ×16) with Poisson noise ($\lambda = 0.05$), results on the FFHQ (Karras et al., 2019) $256 \times 256$ dataset.

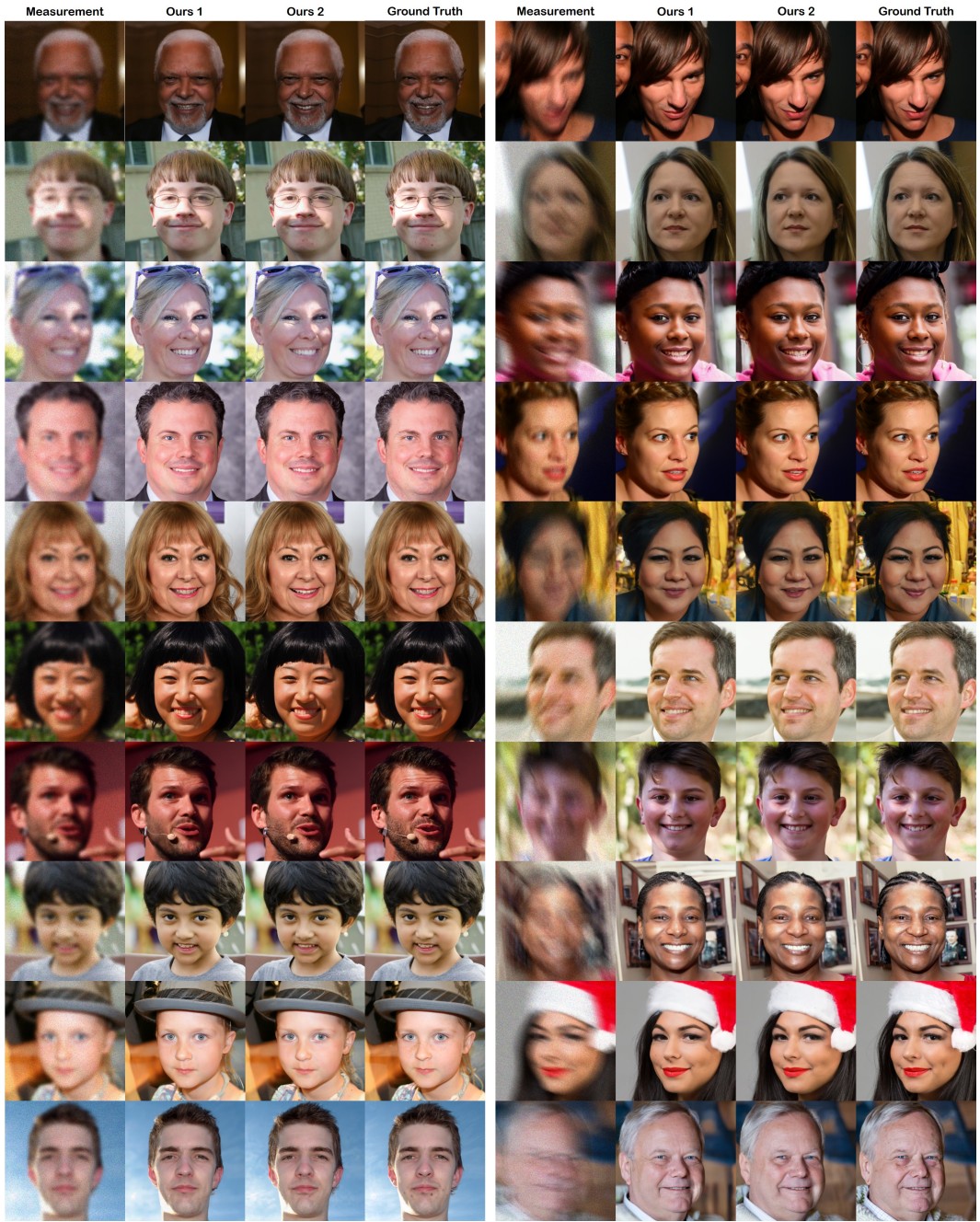

Figure 19: Deblurring results with Poisson noise ($\lambda = 1.0$) (Left Gaussian, Right motion) on the FFHQ (Karras et al., 2019) $256 \times 256$ dataset.

