# OpenReview forum: "Diffusion Posterior Sampling for General Noisy Inverse Problems"
_ICLR.cc/2023/Conference — ICLR 2023 notable top 25%_

### Official Review · Reviewer_ooZZ · 2022-10-17

**Confidence:** 3
**Correctness:** 3
**Technical Novelty And Significance:** 2
**Empirical Novelty And Significance:** 3
**Recommendation:** 6

**Clarity, Quality, Novelty And Reproducibility:**

The paper is well written. The authors correctly place their work in the with respect to existing bodies of work. Sufficient details are given in my opinion to reproduce the results, not to mention that the authors also provide a link to the code. The novelty is somewhat limited, the authors propose more or less an application of diffusion probabilistic models.

**Strength And Weaknesses:**

The main selling point of the paper is that it proposes a rather straightforward framework for solving general Bayesian noisy inverse problems using diffusion probabilistic models. The numerical results show that by using the framework one can obtain high quality results for a diverse set of problems. It is worth mentioning that the proposed method works on top of existing diffusion probabilistic models.

Its main weakness is that overall, it brings little to the table in terms of new knowledge. The paper is interesting, however, and with the risk of oversimplifying a bit the contributions, it is a collection of existing models and/or techniques. While the use of the Laplace approximation is clever in allowing a straightforward approximation of the likelihood term in the reverse diffusion, it is arguably a standard technique in a statistician and ML practitioner's toolbox.

**Summary Of The Paper:**

The paper proposes a framework for general Bayesian noisy inverse problems using diffusion probabilistic models. The unknown data distribution is expressed as a posterior distribution. The updated reverse diffusion process now involves a generally intractable likelihood term. The authors propose the use of Laplace approximation of the likelihood term to circumvent the intractability issue. Results are presented that show the usefulness of the proposed framework.

**Summary Of The Review:**

The paper is overall interesting, the issue is the novelty factor which is low. The paper could still appeal to a sufficiently large audience. I'm not convinced on accepting the paper, but I'm not convinced on rejecting it either. I'm looking forward to reading the rebuttal on the novelty factor.

---

> ### Author Response · Authors · 2022-11-13
> **Official Comment by Paper1619 Authors**
>
> > **W&Q1**. Its main weakness is that overall, it brings little to the table in terms of new knowledge. The paper is interesting, however, and with the risk of oversimplifying a bit the contributions, it is a collection of existing models and/or techniques. While the use of the Laplace approximation is clever in allowing a straightforward approximation of the likelihood term in the reverse diffusion, it is arguably a standard technique in a statistician and ML practitioner's toolbox.
>
> Please see the general comments. In the revised manuscript, we have devoted our best efforts to simplify the derivation, and to make the theorem more useful and understandable, without arbitrary assumptions that are needed. As a result, we have proved an approximation bound that is used in our method, which precisely shows the condition in which the approximation is tight. Our theoretical result now provides better insight on when DPS would provide a good approximation.
>
> Moreover, while the contribution of DPS in terms of adding new “knowledge” may be limited, we would like to emphasize that DPS is a huge leap in terms of widening the applicability of diffusion models in inverse imaging. Specifically, we are the first to show that one can solve for any arbitrary inverse problems that are differentiable with unprecedented quality, even when the measurements are highly noisy. All you need is a well-trained score function. In our humble opinion, we believe that our method will have a great impact in taking generative model-based inverse problem solvers to another level.

---

> > ### Comment · Reviewer_ooZZ · 2022-11-19
> > **Read the authors rebuttal**
> >
> > I would like to thank the authors for their effort in providing answers to the questions raised by all the reviewers. I read their rebuttals and I'm better convinced by the quality of the paper.
> >
> > I just noticed a small error in algorithms 1 and 2: the iteration should be from i=N to 1 and not i=N-1 to 0, computing $x_{i-1}$ at the last iteration would amount to computing $x_{-1}$ where we want to have $x_{0}$.

---

### Official Review · Reviewer_dAym · 2022-10-18

**Confidence:** 4
**Correctness:** 4
**Technical Novelty And Significance:** 3
**Empirical Novelty And Significance:** 3
**Recommendation:** 8

**Clarity, Quality, Novelty And Reproducibility:**

## Clarity

The paper is straightforward and so very clear.

## Quality

This is good work.

## Novelty

While the denoising diffusion models are not new, their application to inverse problems is still work in progress. The proposed framework is a clear contribution on this subject and is very novel.

## Reproducibility

All the work is reproducible. The paper is very clear on the parameters, context and networks. Moreover the code is available.

**Strength And Weaknesses:**

## Strength

The proposed methods has several interesting contributions,
1. The forward process is easy to implement and don't ask for heavy operations like SVD.
2. All the framework is well motivated and the explanation are clear.
3. Results are impressive compared to state-of-art.
4. The full framework seems very extensible and then may interest a large part of image processing community.

## Weaknesses

There is no clear weakness. My only regrets is there is no comparison against more classical methods (e.g image denoising with sparse prior and ADMM as solver).

**Summary Of The Paper:**

This paper proposes a framework based on denoising diffusion model for solving inverse problem in image processing. The main contribution is the characterization of the forward process for non-linear models which include many well known image formation models. The proposed method has the specificity to both be easier to use than previous models (e.g no SVD) and relies on automatic differentiation for complex computation. The results shows great improvement compare to others methods based on diffusion models.

**Summary Of The Review:**

This paper proposes a denoising diffusion process for inverse problems. The main contribution is on the forward process that is well described and easy to implement. The experiments show promising results for image processing problems.

---

> ### Author Response · Authors · 2022-11-13
> **Official Comment by Paper1619 Authors**
>
> > **Comment**. There is no clear weakness. My only regrets is there is no comparison against more classical methods (e.g image denoising with sparse prior and ADMM as solver).
>
> Thank you for your encouraging comments. Indeed, we agree that including comparisons with more classical methods would be beneficial. Per your suggestion, we include comparison with the TV sparsity prior, solved with ADMM. Please see modified Table.1,2,5,6.

---

> > ### Comment · Reviewer_dAym · 2022-11-15
> > **Official Comment by Paper1619 Authors**
> >
> > Thank you for the additional results. They are really interesting and show there are still rooms for improvements.

---

### Official Review · Reviewer_VKPQ · 2022-10-23

**Confidence:** 5
**Correctness:** 3
**Technical Novelty And Significance:** 3
**Empirical Novelty And Significance:** 3
**Recommendation:** 6

**Clarity, Quality, Novelty And Reproducibility:**

The paper is well written for the most parts. Some steps of derivations need more clarity.

**Strength And Weaknesses:**

Strength
- extensive evaluation results and good improvements over SOTA methods
- addressing an important and timely problem
- simple approach but seems effective
- well written for the most parts


Weakness
- The main result in Theorem 1 includes quite strict assumptions that may not hold true for distributions with scores with order beyond linear such as Gaussian. This can really limit the scope of Theorem 1.

- The result in Theorem 1 seems to be a simple first order approximation, but it’s been derived in a complex manner. The Laplace approximation to end up with the result in Theorem 1 seems not necessary.

- The approximation can be very loose for large noise levels at the later steps of the diffusion process because MMSE estimator will be quite far from the ground-truth x_0


Questions and comments
- The assumptions in Theorem 1 seem to be quite restrictive, which does limit the scope of the results and they don’t even include Gaussian distribution. First of all, what's the intuitive meaning of the assumptions? Second, the authors need to elaborate when these assumptions are satisfied and how limited the scope of the distributions satisfying these assumptions would be.

- in section 3.2. it is not quite clear, how p(y|x_0) is related to p(y|\hat{x}_0)? This seems to be coming from proposition 1, but it's not clear since x_0 and \hat{x}_0 could have completely different distributions. Is that exact or is there an additional hidden assumption here? This needs to be clarified in the paper.

- The choice of step size seems to be very specific in this work. Did you try other step sizes such as the constant or diminishing step size rule? it seems that the step size rule adopted here is the key to suppress noise? what if one uses the step size coming from eq. 12, namely 1/sigma^2? an ablation on the choice of step size would be needed to clarify the sensitivity of the algorithm to step size.

- better to move tables/algorithms to the top part of each page
- page 3, last paragraph, line 6, is there a typo in p(x|y)? shouldn’t it be p(y|x)?
- page 3, last line, remove “the” from “.... both lines of the works …”



**Summary Of The Paper:**

this submission deals with posterior sampling from diffusion models for inverse problems. For general inverse problems with nonlinearity and measurement noise, sampling the conditional posterior needs the conditional score \log \nablap(y|x_t), that is very complicated and can be quite far from Gaussian, at a generic time step t. The idea in this submission is to (first-order) approximate it around the optimal MMSE estimate at each time t, namely \hat{x}_0. Several experiments, provided for linear and nonlinear inverse tasks, show significant quality (FID & LPIPS) improvement over SOTA with this simple approximation.

**Summary Of The Review:**

This paper deals with an important and timely problem. The idea is original, and the experimental results are solid. However, the main result in Theorem 1 seems to be quite restrictive to a small class of distributions that limits the scope of the theorem.

---

> ### Author Response · Authors · 2022-11-13
> **Official Comment by Paper1619 Authors (1/2)**
>
> We would like to thank the reviewer for the constructive comments and the thorough feedback. For point-to-point response, see below.
>
> > **W&Q1,2**. The main result in Theorem 1 includes quite strict assumptions that may not hold true for distributions with scores with order beyond linear such as Gaussian. This can really limit the scope of Theorem 1. The assumptions in Theorem 1 seem to be quite restrictive, which does limit the scope of the results and they don’t even include Gaussian distribution. First of all, what's the intuitive meaning of the assumptions? Second, the authors need to elaborate when these assumptions are satisfied and how limited the scope of the distributions satisfying these assumptions would be. The result in Theorem 1 seems to be a simple first order approximation, but it’s been derived in a complex manner. The Laplace approximation to end up with the result in Theorem 1 seems not necessary.
>
> Thank you for your constructive and inspiring comment. Our previous theorem involved rather unrealistic assumptions that precluded clear understanding of the intuitive meanings. However, as stated in the general comment 1, we have now rederived our main result without any explicit assumptions. Rather, we proved that there exists an upper bound in the error for the approximation proposed in our work, where we can clearly see that the error approaches 0 when the noise level of the measurement increases. Our theorem matches the empirical results in our method shows particular robustness with high level of noise in the measurement.
>
> > **W&Q3**. The approximation can be very loose for large noise levels at the later steps of the diffusion process because MMSE estimator will be quite far from the ground-truth $\boldsymbol{x}_0$
>
> Generally, at higher noise levels (in the context of diffusion models; $t \sim T$) it is much harder for the score function to perfectly denoise the extremely noisy images.  However, note that the accuracy of the denoisers (score functions) gradually gets better as we reach the end of the process, thereby correcting the errors in the initial steps. Indeed, we believe that such coarse-to-fine generation procedure is one of the main reasons why diffusion models show such strong performance in inverse imaging.
>
> > **W&Q4**. In section 3.2. it is not quite clear, how $p(\boldsymbol{y}|\boldsymbol{x}_0)$ is related to $p(\boldsymbol{y}|\hat{\boldsymbol{x}}_0)$? This seems to be coming from proposition 1, but it's not clear since $\boldsymbol{x}_0$ and $\hat{\boldsymbol{x}}_0$ could have completely different distributions. Is that exact or is there an additional hidden assumption here? This needs to be clarified in the paper.
>
> Thank you for your constructive and inspiring comment. Our previous theorem involved rather unrealistic assumptions that may lead to the confusion. In regard to the specific question, $\hat{\boldsymbol{x}}_0$  is the posterior mean at the intermediate steps, which is only being used as the plug-in $\boldsymbol{x}_0 \gets \hat{\boldsymbol{x}}_0$ during the sampling.
>
> > **W&Q5**. The choice of step size seems to be very specific in this work. Did you try other step sizes such as the constant or diminishing step size rule? it seems that the step size rule adopted here is the key to suppress noise? what if one uses the step size coming from eq. 12, namely $1/\sigma^2$? an ablation on the choice of step size would be needed to clarify the sensitivity of the algorithm to step size.
>
> When choosing the step size, we opted for maximal simplicity, while being effective. While the choices presented in Appendix D.1. may seem very specific and complicated, the implementation of such step sizes is very simple. In essence, we are taking static step sizes with the un-squared norm of the residuals. (See our implementation: [line of code](https://github.com/DPS2022/diffusion-posterior-sampling/blob/main/guided_diffusion/condition_methods.py#L86)). Having said that, we note that our scheme is robust to noise regardless of step size, regardless of the scheduling. However, we do observe quite a difference in terms of reconstruction quality, which we elaborate in the following ablation study.
>
> 1. Linearly decaying steps: $\zeta_t = \zeta_{init} \times (1 - \frac{t}{N})$
>
> 2. Exponentially decaying steps: $\zeta_t = \zeta_{init} \times \gamma^t, \gamma = 0.99$
>
> 3. Directly using the step size in eq.(12) $\propto 1/\sigma^2$
>
> | Strategy      | constant   | constant     | linear decay | linear decay | exp decay | exp decay |
> |---------------|------------|--------------|--------------|--------------|-----------|-----------|
> | Initial value | **1.0 (ours)** | $1/\sigma^2$ | 0.3          | 1.0          | 0.3       | 1.0       |
> | LPIPS         | **0.247**      | 0.727        | 0.287        | 0.251        | 0.421     | 0.442     |

---

> > ### Author Response · Authors · 2022-11-13
> > **Official Comment by Paper1619 Authors (2/2)**
> >
> > From the results presented in the above table, we verify that our choice of step size yields the best result despite the simplicity. We have included Appendix C.3. to give details on this experiment, together with visual results where we clearly see the superiority of the proposed method.
> >
> > > **MC1.** Better to move tables/algorithms to the top part of each page.
> >
> > Done, except for Figure 5, and Table 3.,4., which is placed in a sub-window within the paragraph.
> >
> > > **MC2, 3.** page 3, last paragraph, line 6, is there a typo in $p(x|y)$? shouldn’t it be $p(y|x)$? Page 3, last line, remove "the" from "... both lines of the works ... ".
> >
> > Fixed.

---

### Official Review · Reviewer_3uLY · 2022-10-23

**Confidence:** 4
**Correctness:** 3
**Technical Novelty And Significance:** 3
**Empirical Novelty And Significance:** 2
**Recommendation:** 8

**Clarity, Quality, Novelty And Reproducibility:**

*Clarity:* well-written and easy to follow
*Quality:* the overall quality of the manuscript is high
*Novelty:* the work is novel
*Reproducibility:* looks like it could be reproduced


**Strength And Weaknesses:**

*Strength*:
1) The manuscript is well-written and easy to follow.
2) The figures are illustrative and useful for readers to understand the problem and the approach.
3) The proposed DPS is novel and provides a new way to compute $p_t(y|x^t)$
4) Clear experiments with sharp results

*Weakness & Questions*:
1. The connection between Proposition 2 and Theorem 1 is not clear in the text. Please provide more explanation on how Proposition 2 helps derive Theorem 1. I notice the proof in the appendix, what I suggest is to include some high-level explanation.
2. Fig. 3 is very illustrative. However, it can mislead readers to interpret that DPS sticks to each manifold $\mathcal{M}_n$ perfectly, which I don't think is true as multiple layers of approximation are applied. I suggest the authors emphasize in the caption that this is just a conceptual illustration of DPS.
3. In the experiments, performance is only evaluated by using perception loss rather than PSNR/SNR/SSIM. Is it because diffusion models generally have bad performance under these metrics? It will be interesting to the PSNR/SSIM comparison because reconstruction accuracy is much more important than perceptual quality in imaging inverse problems.
4. What is the denoiser of PnP-ADMM? If it is not a deep denoiser, I suggest comparing DPS with a deep denoiser.

*Minors:
1. Eq. 11 and Alg. 1 & 2 are not consistent.
2. Please clarify the definition of $\simeq$.


**Summary Of The Paper:**

This paper considers an important problem in denoising diffusion models (DPM), that is, how to accurately conduct posterior sampling given a pre-learned score function and an inverse problem. Previous works address this problem by circumventing the calculation of $p(y|x^t)$ and resorting to projections onto the measurement subspace. However, these works either achieve unsatisfactory performance or are not elegant in terms of theoretical interpretation. Different from prior methods, diffusion posterior sampling (DPS) proposes to approximately compute $p_t(y|x^t)$ for each diffused image $x^t$, and conduct posterior sampling using the discretized version of the following equation
$$dx=[-\frac{\beta(t)}{2}x-\beta(t)(\nabla_{x_t}\text{log} p_t(x_t) + \nabla_{x_t} \text{log} p_t(y|x_t))]dt + \sqrt{\beta{t}} d\bar{w}$$

In summary, the contribution of this work is:
1. It proposes a feasible approach to compute $p_t(y|x_t)$ by using Tweedie's formula and Laplace's method. In particular,
$$\nabla_{x_t} \text{log} p_t(y|x^t) \sim \nabla_{x_t} \text{log} p_t(y|\hat{x}_0),\quad\text{where}\quad \hat{x}_0 = \text{TweedieFormula}(x_t)$$
2. It experimentally shows that DPS achieves better results than the state-of-the-art algorithms on multiple inverse problems

**Summary Of The Review:**

I think this paper makes concrete contributions to the literature on diffusion models with application to inverse problems. The method DPS is novel, and the theoretical interpretation in the work is insightful.

---

> ### Author Response · Authors · 2022-11-13
> **Official Comment by Paper1619 Authors (1/2)**
>
> Thank you for your detailed feedback and the encourging comments. Please see the detailed response below.
>
> > **W&Q1**. The connection between Proposition 2 and Theorem 1 is not clear in the text. Please provide more explanation on how Proposition 2 helps derive Theorem 1. I notice the proof in the appendix, what I suggest is to include some high-level explanation.
>
> Please see general comment 1 along with the modified manuscript. When re-deriving the main theorem, we have made sure that the main text contains high-level explanation that leads to the method.
>
> > **W&Q2**. Fig. 3 is very illustrative. However, it can mislead readers to interpret that DPS sticks to each manifold perfectly, which I don't think is true as multiple layers of approximation are applied. I suggest the authors emphasize in the caption that this is just a conceptual illustration of DPS.
>
> As the reviewer pointed out, we can guarantee neither 1) a single-step reverse diffusion step will arrive exactly at the next level of the manifold, and 2) DPS will evolve exactly sticking to the manifold. As the main point of Figure 3 was to illustrate why DPS would be desirable in noisy inverse problem settings in a high-level context, we have made clear in the caption figure that the figure is a **conceptual illustration**. See modified figure 3.
>
> > **W&Q3**. In the experiments, performance is only evaluated by using perception loss rather than PSNR/SNR/SSIM. Is it because diffusion models generally have bad performance under these metrics? It will be interesting to the PSNR/SSIM comparison because reconstruction accuracy is much more important than perceptual quality in imaging inverse problems.
>
> We mainly chose perception metrics as the degradations used in the manuscript are rather aggressive. When this is the case, the range of feasible reconstructions is quite large such that it is much less meaningful to quantify standard metrics such as PSNR/SSIM (i.e. it is less meaningful to consider the “accuracy” of reconstruction). In this regime, the reconstruction that will score the highest in terms of PSNR/SSIM is the mean of the feasible reconstructions, which is typically blurry and omits meaningful details. On the other hand, our method performs approximate posterior sampling, thus providing diverse samples (that are perceptually very close to the ground truth) from the posterior distribution. When compared with methods that produce blurry reconstructions (e.g. PnP-ADMM), while it is clear qualitatively that DPS provides higher quality (e.g. preserving high-frequency detail) samples, PnP-ADMM more often than not scores higher on PSNR/SSIM.
>
> Furthermore, for cases such as box-type inpainting, where **any** semantically relevant inpainting results would suffice for “good” reconstructions, measuring the pixel-wise deviation from the ground truth has very little value. At best, PSNR/SSIM would measure the performance of “denoising”, as even the measured pixels are contaminated with noise.
>
> Having said that, we agree that clearly stating the performance in terms of more standard metrics would be beneficial. We have included Table 6 and Table 7, focusing on these metrics. For the reviewer’s convenience, we include the table here.
>
> #### Table 6. Quantitative evaluation (PSNR, SSIM) of solving linear inverse problems on FFHQ 256x256 1k validation dataset. **Bold**: best, *underline*: second best.
>
> |           | SR(x4)    | SR(x4)    | Inpaint(box) | Inpaint(box) | Inpaint(random) | Inpaint(random) | Deblur (gauss) | Deblur (gauss) | Deblur (motion) | Deblur (motion) |
> |-----------|-----------|-----------|--------------|--------------|-----------------|-----------------|----------------|----------------|-----------------|-----------------|
> | Method    | PSNR      | SSIM      | PSNR         | SSIM         | PSNR            | SSIM            | PSNR           | SSIM           | PSNR            | SSIM            |
> | **DPS(ours)** | _25.67_   | _0.852_   | **22.47**    | **0.873**    | **25.23**       | **0.851**       | _24.25_        | _0.811_        | **24.92**       | **0.859**       |
> | DDRM [1]      | 25.36     | 0.835     | _22.24_      | _0.869_      | 9.19            | 0.319           | 23.36          | 0.767          | -               | -               |
> | MCG [2]      | 20.05     | 0.559     | 19.97        | 0.703        | _21.57_         | _0.751_         | 6.72           | 0.051          | 6.72            | 0.055           |
> | PnP-ADMM [3] | **26.55** | **0.865** | 11.65        | 0.642        | 8.41            | 0.325           | **24.93**      | **0.812**      | _24.65_         | _0.825_         |
> | Score-SDE [4] | 17.62     | 0.617     | 18.51        | 0.678        | 13.52           | 0.437           | 7.12           | 0.109          | 6.58            | 0.102           |
> | ADMM-TV   | 23.86     | 0.803     | 17.81        | 0.814        | 22.03           | 0.784           | 22.37          | 0.801          | 21.36           | 0.758           |

---

> > ### Author Response · Authors · 2022-11-13
> > **Official Comment by Paper1619 Authors (2/2)**
> >
> > #### Table 7. Quantitative evaluation (PSNR, SSIM) of solving linear inverse problems on ImageNet 256x256 1k validation dataset. **Bold**: best, *underline*: second best.
> >
> > |           | SR(x4)    | SR(x4)    | Inpaint(box) | Inpaint(box) | Inpaint(random) | Inpaint(random) | Deblur (gauss) | Deblur (gauss) | Deblur (motion) | Deblur (motion) |
> > |-----------|-----------|-----------|--------------|--------------|-----------------|-----------------|----------------|----------------|-----------------|-----------------|
> > | Method    | PSNR      | SSIM      | PSNR         | SSIM         | PSNR            | SSIM            | PSNR           | SSIM           | PSNR            | SSIM            |
> > | **DPS(ours)** | _23.87_   | _0.781_   | **18.90**    | _0.794_      | **22.20**       | **0.739**       | _21.97_        | **0.706**      | _20.55_         | _0.634_         |
> > | DDRM [1]      | **24.96** | **0.790** | _18.66_      | **0.814**    | 14.29           | 0.403           | **22.73**      | _0.705_        | -               | -               |
> > | MCG  [2]     | 13.39     | 0.227     | 17.36        | 0.633        | _19.03_         | _0.546_         | 16.32          | 0.441          | 5.89            | 0.037           |
> > | PnP-ADMM [3]  | 23.75     | 0.761     | 12.70        | 0.657        | 8.39            | 0.300           | 21.81          | 0.669          | **21.98**       | **0.702**       |
> > | Score-SDE [4] | 12.25     | 0.256     | 16.48        | 0.612        | 18.62           | 0.517           | 15.97          | 0.436          | 7.21            | 0.120           |
> > | ADMM-TV   | 22.17     | 0.679     | 17.96        | 0.785        | 20.96           | 0.676           | 19.99          | 0.634          | 20.79           | 0.677           |
> >
> > > **W&Q4**. What is the denoiser of PnP-ADMM? If it is not a deep denoiser, I suggest comparing DPS with a deep denoiser.
> >
> > The denoiser for PnP-ADMM is already set to DnCNN, as stated in Appendix D.2. We have made this clear in the main text to avoid confusion.
> >
> > > **MC1**. Eq. 11 and Alg. 1 & 2 are not consistent.
> >
> > Fixed.
> >
> > > **MC2**. Please clarify the definition of $\simeq$
> >
> > There are multiple sources of approximations used in our work, which we simply denote as $\simeq$ for simplicity. Here, we list the meaning of $\simeq$ in each context.
> >
> > 1. Score function estimation error
> >
> > $s_{\theta^*} \simeq \nabla_{\boldsymbol{x}_t} \log p(\boldsymbol{x}_t)$ has an approximation error coming from 1) optimization error, and the 2) parameterization error of the neural network. Generally, the error grows larger as $t \rightarrow T$, and closes the gap as $t \rightarrow 0$.
> > The approximation is used in 1) solving reverse SDE numerically, 2) producing $\hat{\boldsymbol{x}}_0$ at the intermediate timesteps (eq. (11)).
> >
> > 2. Jensen gap
> >
> > We have quantified the Jensen gap by deriving the upper bound for our measurement model. Please see modified Theorem 1.
> >
> > We have modified our manuscript to be clear on the meaning of $\simeq$.
> >
> > ---
> >
> > ### References.
> > [1] Kawar, Bahjat, et al. "Denoising diffusion restoration models." NeurIPS (2022).
> >
> > [2] Chung, Hyungjin, et al. "Improving Diffusion Models for Inverse Problems using Manifold Constraints." NeurIPS (2022).
> >
> > [3] Chan, Stanley H., Xiran Wang, and Omar A. Elgendy. "Plug-and-play ADMM for image restoration: Fixed-point convergence and applications." IEEE Transactions on Computational Imaging 3.1 (2016): 84-98.
> >
> > [4] Song, Yang, et al. "Score-based generative modeling through stochastic differential equations." ICLR (2021).

---

> > > ### Comment · Reviewer_3uLY · 2022-11-14
> > > **Thank you for the response.**
> > >
> > > Your response has clarified my concerns.

---

### Public Comment · ~Yiwei_Guo1 · 2022-11-12
**Eq.(21) in the proof seems to confuse "global maxima" with "expectation"**

The paper claims to find a global maxima of $p(x_0 | x_t)$, stated in Eq.(21). The provided derivation is done by taking gradient w.r.t $x_t$ on Eq.(20). This is not correspondent to seeking the maxima of $p(x_0 | x_t)$, because one should take gradient of $x_0$ instead. By Tweedie's formula as the authors have mentioned, the Eq.(21) actually shows that $\hat x_0$ is an **expectation** rather than the **global maxima** of posterior.

There is an intuitive explanation why it cannot be global maxima. Think of $x_0$ takes $-1$ or $1$ with equal probability, then $p(x_0 | x_t=0)$ must be an even function, i.e. $\nabla _{x_t} \log p(x_0 | x_t = 0) = 0$. Therefore, $\hat x_0 = 0$. But obviously the posterior has two modes on the left and right side of 0. In other words, Eq.(21) implies there is a unique global maxima, but in reality this is not always the case. Instead, there would be a unique expectation all the time.

This probable mistake, if it is, will lead to the break of logical chain. Laplace's method (Proposition 2) requires that function $f(\cdot)$ has a global maxima, so it's hard to be applied to the posterior distribution. Meanwhile, Proposition 2 is necessary in the proof of the paper. As a result, the theoretical correctness seems a little bit doubtful.

---

> ### Author Response · Authors · 2022-11-12
> **Thanks for your comments, which help us to significantly simplify our derivation.**
>
> Thanks for the careful reading and for pointing out the issues we had not noticed before.
>
> Inspired by your comments,  we now provide a rigorous proof that  $\hat x_0$ from Tweedie's formula is indeed the posterior mean as you suspected.  That said, we would also like to assure the reviewer that this observation also leads to a much simpler derivation of our main results without even using the Laplace approximation, and helps us to get rid of other restricted assumptions (such as  Hessian of the score is approximately zero) that were pointed out by other reviewers.
>
> More specifically,  we found that our method is indeed using the approximation $E[f(x_0)]\simeq f[E(x_0)]$ where $E[]$ is the posterior mean.  In fact, the approximation error between the two, ie. $J(f, P_{x_0|x_t}) = E[f(x_0)]- f[E(x_0)]$ is called the Jensen gap, where $f$ is  either convex or non-convex.  Therefore, in the revised paper, we investigate the conditions to reduce the Jensen gap, and to our expectation,  for the noisy inverse problem, we can show that the Jensen gap approaches zero as the noise level increases. This theoretical proof also justifies why our method works well for noisy inverse problems. Furthermore, we would like to point out that our new derivation does not require any other assumption other than the noise levels, which significantly simplifies the derivation and clarifies our contributions.
>
> Thanks again for your interest in our work and very careful reading. We really appreciate your meticulous comments which have significantly improved the quality of our paper.

---

### Author Response · Authors · 2022-11-13
**Official Comment by Paper1619 Authors**

We would like to thank the reviewers for their constructive and thorough reviews. We are encouraged that the reviewers think that our paper is “well written, easy to follow, novel, with clear and sharp experiments” (3uLY), the paper “addresses an important and timely problem, and the method is simple but effective” (VKPQ), “proposes a clear contribution on the subject and is very novel” (dAym), and “straightforward and obtains high quality results on diverse set of problems” (ooZZ).

Many of the reviewers’ concerns were pointed at the unclear exposition of the underlying theory that derives the proposed method. In fact, Yiwei Guo kindly left a public comment that \hatx0 may be the posterior mean. Inspired by the comment, we have re-derived our result. Interesting enough, derivation of the method for DPS now became clearer, without needing specific assumptions that the original theorem had, and being able to the quantify the approximation error.

## 1. Fix and elaborate theoretical claims

In essence, our new theorem states that $\log p(\boldsymbol{y}|\boldsymbol{x}_t) \simeq \log p(\boldsymbol{y}|\hat{\boldsymbol{x}}_0)$, and we can quantify the approximation error with the Jensen gap $\mathcal{J}(f, p(\boldsymbol{x}_0|\boldsymbol{x}_t)) := \mathbb{E}[p(\boldsymbol{y}|\boldsymbol{x}_0)] – p(\boldsymbol{y}|\hat{\boldsymbol{x}}_0)$. By specifying the upper bound on the Jensen gap, we are able to see that the gap approaches zero as the noise level increases, explaining the strength of the proposed method in highly noisy inverse problems.

## 2. Quantitative evaluation

Quantitative evaluations presented in Table 1,2 were focused on perception metrics – LPIPS/FID, and not on standard distortion metrics such as PSNR/SSIM. We further report on the standard distortion metrics to be as clear as possible in the presentation of our empirical results. See Table 6,7 in Appendix.

## 3. Ablation studies

An ablation study comparing the different choices for the step sizes was included. Among many different choices, we show that the proposed step size is indeed the most effective while being simple.

## 4. Comparison against classical methods

One of the reviewers suggested that the paper would be stronger with comparisons against more classic methods (image denoising with sparse prior and ADMM as a solver) as comparison. Per the suggestion, we have included ADMM-TV as another method to compare against.

---

> ### Comment · Reviewer_dAym · 2022-11-30
> **Final version of the paper**
>
> Many thanks for this works. I think this paper is a great contribution to image processing that really deserve to be publish at ICLR. I stay with a score of 8.

---

### Decision · Program_Chairs · 2023-01-20

**Decision:**

Accept: notable-top-25%

**Justification For Why Not Higher Score:**

Using standard approximation techniques, the paper provides an incremental improvement of an existing diffusion model-based inverse problem pipeline.

**Justification For Why Not Lower Score:**

Identifying a key place for improvement under an existing inverse-problem pipeline, the proposed solution is simple and effective.

**Metareview: Summary, Strengths And Weaknesses:**

The paper shows how to leverage a pretrained diffusion model as the prior to solve general noisy inverse problems. Combining a pretrained score estimator and an estimator of the gradient of the log likelihood, it arrives at a reverse diffusion sampler for sampling from the posterior distribution. The key technical contribution is how to use the Laplace approximation to better approximate the intractable log likelihood $\log p(y|x_t)$. Empirical results, involving both linear and nonlinear corruptions, show that the proposed method provides an effective solution for a variety of noisy inverse problems. The main weakness is that assembling a collection of existing techniques, the proposed method does not bring much new knowledge on how to leverage a pretrained diffusion model to solve image inverse problems. All four reviewers recognize the contributions of the paper in identifying a key place for improvement on using pretrained diffusion models for solving image inverse problems and achieving state-of-the-art results.

**Note From Pc:**

if the above contains the word "oral" or "spotlight" please see: "oral" presentation means -> notable-top-5% and "spotlight" means -> notable-top-25%. As stated in our emails, we are disassociating presentation type from AC recommendations